# Conformal Prediction for Ensembles: Improving Efficiency via Score-Based Aggregation

**Eduardo Ochoa Rivera**[*]
Department of Statistics
University of Michigan
Ann Arbor, MI 48104
eochoa@umich.edu

**Yash Patel**[*]
Department of Statistics
University of Michigan
Ann Arbor, MI 48104
yppatel@umich.edu

**Ambuj Tewari**
Department of Statistics
University of Michigan
Ann Arbor, MI 48104
tewaria@umich.edu

## Abstract

Distribution-free uncertainty estimation for ensemble methods is increasingly desirable due to the widening deployment of multi-modal black-box predictive models. Conformal prediction is one approach that avoids making strong distributional assumptions. Methods for conformal aggregation have been proposed for ensembled prediction, where the prediction regions of individual models are merged to retain coverage guarantees while minimizing conservatism. Merging the prediction *regions* directly, however, can miss out on opportunities to further reduce conservatism by exploiting structures present in the conformal *scores*. We, therefore, propose a novel framework that extends the standard scalar formulation of a score function to a multivariate score that produces more efficient prediction regions. We then demonstrate that such a framework can be efficiently leveraged in both classification and predict-then-optimize regression settings downstream and empirically show the advantage over alternate conformal aggregation methods.

## 1   Introduction

Ensemble methods are an oft-used class of statistical modeling techniques due to their ability to reduce variance or improve predictive accuracy [1, 2, 3]. Such methods are increasingly being coupled with complex, black-box models, such as in multi-modal language models [4, 5, 6, 7, 8]. Couplings of this sort are seeing ever-widening deployment in safety-critical settings, such as medicine [9, 10, 11] and robotics [12, 13, 14].

Increasing interest is, therefore, now being placed on quantifying uncertainty for such models [15, 16, 17, 18, 19]. Towards this end, methods of uncertainty quantification have arisen, such as deep ensembles and committee estimation [20, 21, 22]. Such methods, however, sacrifice generality with the imposition of distributional assumptions, motivating the need for distribution-free uncertainty quantification for ensemble methods.

One method for performing distribution-free uncertainty quantification is conformal prediction, which provides a principled framework for producing distribution-free prediction regions with marginal frequentist coverage guarantees [23, 24]. By using conformal prediction on a user-defined score function, prediction regions attain marginal coverage guarantees. While calibration is guaranteed from this procedure, predictive efficiency, i.e., the size of the resulting prediction regions, can be large for poorly chosen score functions.

As a result, methods have arisen to perform conformal model aggregation, which both provide uncertainty estimates of the ensembled predictions and do so in ways as to minimize the prediction region size [25, 26, 27, 28, 29]. While such approaches succeed in reducing the prediction region

---

[*]Denotes alphabetic ordering indicating equal contributions.

39th Conference on Neural Information Processing Systems (NeurIPS 2025).

$$\mathcal{S} = \{(s_1(x_i), s_2(x_i))\}_{i=1}^{N_C} \qquad \Pi_{u_m}(\mathcal{S}) \qquad \widehat{\mathcal{Q}} = \cap_{m=1}^{M} H(u_m, \widehat{q}_m)$$

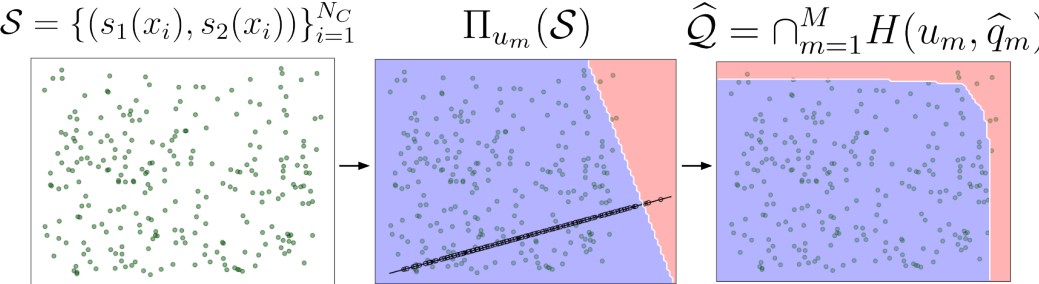

Figure 1: CSA provides a principled extension to the standard conformal prediction pipeline by leveraging ideas from higher-dimensional quantile regression to define quantile envelopes $\widehat{\mathcal{Q}}$ instead of scalar quantiles $\widehat{q}$. It does so by evaluating a collection of score functions (here $s_1$ and $s_2$) over the calibration dataset to define $\mathcal{S}$, finding quantiles $\{\widehat{q}_m\}$ over a set of projection directions $\{u_m\}$, and taking $\widehat{\mathcal{Q}}$ to be the intersection of the resulting half-planes $H(u_m, \widehat{q}_m)$. These quantile envelopes result in more informative prediction regions that can be used in downstream tasks.

size over naive aggregation, they all aggregate the *separately conformalized* prediction regions of the predictors in the ensemble. In doing so, they forgo the possibility of automatically leveraging shared structure amongst the scores of the individual predictors, resulting in conservative prediction regions.

We instead propose to perform aggregation in *score space* by extending traditional conformal prediction to consider a multivariate score function and defining prediction regions using "quantile envelopes" in place of scalar quantiles. Doing so enables efficient, data-driven, automated conformal model aggregation. We demonstrate that this formulation retains the desired distribution-free coverage guarantees typical of standard conformal prediction and that the resulting prediction regions can be used efficiently in both classification and regression settings. Our contributions are:

- Providing a multivariate extension to conformal prediction, dubbed "conformal score aggregation" (CSA), that leverages quantile envelopes to enable data-driven, informative uncertainty estimation for model ensembles while retaining coverage guarantees.

- Demonstrating how the prediction regions resulting from CSA can be efficiently leveraged in downstream predict-then-optimize regression tasks.

- Demonstrating the empirical improvement of the CSA framework over alternate conformal aggregation strategies across classification and regression settings.

## 2 Background

### 2.1 Conformal Prediction

Coverage guarantees of uncertainty quantification methods generally rely on distributional assumptions, often via asymptotics or explicit specification. To alleviate the need for such restrictive assumptions, interest in finite-sample, distribution-free uncertainty quantification methods has risen. Conformal prediction is one such method [23, 24].

Conformal prediction serves as a wrapper around such predictors, producing prediction regions $\mathcal{C}(x)$ that have formal guarantees of the form $\mathcal{P}_{X,Y}(Y \notin \mathcal{C}(X)) \le \alpha$ for some prespecified level $\alpha$. To achieve this, "split conformal" partitions the dataset $\mathcal{D} = \{(x_i, y_i)\}_{i=1}^{N}$ into a training set $\mathcal{D}_T$ and a calibration set $\mathcal{D}_C$. The former serves as the data used to fit $\widehat{f}$. Users of conformal prediction must then design a "score function" $s(x, y)$, which should quantify "test error", often in a domain-specific manner. For instance, a simple score function for a regression setting would be $s(x, y) = \|\widehat{f}(x) - y\|$. This score function is then evaluated across the calibration set to define $\mathcal{S}_C = \{s(x, y) \mid (x, y) \in \mathcal{D}_C\}$. For a desired coverage of $1 - \alpha$, we then take $\widehat{q}$ to be the $\lceil(|\mathcal{D}_C| + 1)(1 - \alpha)\rceil/|\mathcal{D}_C|$ quantile of $\mathcal{S}_C$, with which prediction regions for future test queries $x$ can be defined as $\mathcal{C}(x) = \{y \mid s(x, y) \le \widehat{q}\}$. Under the exchangeability of the score of a test point $s(X', Y')$ with $\mathcal{S}_C$, we have the desired *finite-sample* probabilistic guarantee that $1 - \alpha \le \mathcal{P}_{X',Y'}(Y' \in \mathcal{C}(X'))$.

While this guarantee holds for any $s(x, y)$, the informativeness of the resulting prediction regions, quantified as the inverse expected Lebesgue measure across $X$, i.e. $(\mathbb{E}[\mathcal{L}(\mathcal{C}(X))])^{-1}$, is intimately tied to its specification [24]. Thus, much of the challenge of conformal prediction relates to choosing a score function that retains coverage while minimizing region size.

## 2.2 Quantile Envelopes

Generalizations of quantiles have a long history in statistics [30, 31]. Unlike univariate data, multivariate data do not lend itself to an unambiguous definition of a quantile, as there is no canonical ordering in higher dimensional spaces. The notion of a "directional quantile" for a random variable $X \in \mathbb{R}^n$ can, however, be directly defined given some direction $u \in \mathcal{S}^{n-1}$, namely as $Q(X, \alpha, u) = \inf\{q \in \mathbb{R} : \mathcal{P}(u^\top X \le q) \ge \alpha\}$ [32, 33, 34]. When there is no ambiguity, we just denote it as $Q(\alpha, u)$. For any given $u$, notice the choice of quantile defines a corresponding halfplane $H(u, Q(\alpha, u)) = \{x \in \mathcal{X} : u^\top x \le Q(\alpha, u)\}$. The quantile envelope is then the intersection thereof:

$$D(\alpha) = \bigcap_{u \in \mathcal{S}^{n-1}} H(u, Q(\alpha, u)). \tag{1}$$

Notably, while each individual $H(u, Q(\alpha, u))$ captures $1 - \alpha$ of the points, $D(\alpha)$ does *not*, as it is the intersection thereof and hence captures $< 1 - \alpha$ of the mass. If $1 - \alpha$ combined coverage is sought, a correction, such as Bonferroni adjustment, is used for the individual planes.

## 2.3 Predict-Then-Optimize

In the case of classification, conformal prediction regions simply constitute a subset of the label space, making their direct use by end users straightforward [35]. In high-dimensional regression settings, however, prediction regions become harder to use directly; for this reason, recent works have started shifting focus to using them in their implicit forms.

One such application is [36], where conformal prediction was leveraged in a predict-then-optimize setting. As the name suggests, predict-then-optimize problems are two-stage problems, which take observed contextual information $x$ and predict the parametric specification of a downstream problem of interest $\widehat{c} := g(x)$ with some trained predictor $g$. The final result is then a decision made with this specification, $w^* := \min_w f(w, \widehat{c})$. An example of such a setting is if an optimal labor allocation $w^*$ is sought based on predicted demand $\widehat{c}$ from transactions $x$ in a delivery platform.

While the predicted $\widehat{c}$ is often trusted, this approach is inappropriate in risk-sensitive settings, where misspecification of the map $g : \mathcal{X} \to \mathcal{C}$ could lead to suboptimal decision-making. For this reason, recent interest has been placed on studying a "robust" formulation [37, 38, 39]. Following this line of work, [36] proposed studying $w^*(x) := \min_w \max_{\widehat{c} \in \mathcal{C}(x)} f(w, \widehat{c})$, with $\mathcal{C}(x)$ being produced by conformalizing the predictor $g$.

## 2.4 Related Works

Ensemble methods consist of $K$ predictors $f_k : \mathcal{X}_k \to \mathcal{Y}$; notably, such predictors need not map from the same set of covariates. A naive approach for uncertainty quantification would then be to conformalize the ensembled predictor. That is, for an ensembling algorithm $\mathcal{F} : \mathcal{Y}^K \to \mathcal{Y}$, a score function $s(\mathcal{F}(f_1(x), ..., f_K(x)), y)$ would be defined. Denoting the $\lceil (N_\mathcal{C} + 1)(1 - \alpha) \rceil / N_\mathcal{C}$ quantile of the score distribution over $\mathcal{D}_C$ as $\widehat{q}(\alpha)$, $\mathcal{C}(x) = \{y : s(x, y) \le \widehat{q}(\alpha)\}$ would then be calibrated.

Such an approach, however, lacks some desirable properties. In particular, prediction regions $\mathcal{C}(x)$ should have the quality that, if a particular predictor has less uncertainty in its predictions, as is frequently true of ensemble settings where the predictors span multiple input data modalities, upon routing to that predictor, the corresponding size of the prediction region should be smaller than if it had been routed to a different predictor. While the naive approach does, in principle, support this property, it ultimately relies on defining an *uncertainty-aware* ensembling algorithm $\mathcal{F}$. In its typical form, however, $\mathcal{F}$ simply takes *point predictions* $f_1(x), ..., f_K(x)$ in as input, meaning any uncertainty-awareness would need to be baked in a priori into the definition of $\mathcal{F}$ through domain knowledge of the uncertainties of the predictors $f_1, ..., f_K$, which can seldom be specified precisely, sacrificing the predictive efficiency of $\mathcal{C}(x)$.

Conformal model aggregation, thus, seeks to mitigate these deficiencies by aggregating the prediction regions $\mathcal{C}_1(x), ..., \mathcal{C}_K(x)$ rather than the individual point predictions [25, 27, 28, 29]. While there are several methods in this vein, they can be categorized into one of two general approaches. The first line of work seeks to perform model *selection*, in which a single conformal predictor is selected $\mathcal{C}_{k^*}$, typically based on the criterion of minimizing region size $k^* := \arg\min_k \mathbb{E}[\mathcal{L}(\mathcal{C}_k(X))]$ [27, 28].

Generally, however, methods leveraging the full collection of predictors produce less conservative regions [25, 29]. Such works aggregate the individual prediction regions into a final region by defining $\mathcal{C}(x) := \{y \mid \sum_{k=1}^{K} w_k \mathbb{1}[y \in \mathcal{C}_k(x)] \geq \widehat{a}\}$ for weights $\{w_k\} \in [0, 1]$ such that $\sum_{k=1}^{K} w_k = 1$ and a threshold $\widehat{a}$. Methods then differ in the procedure by which $\{w_k\}$ and $\widehat{a}$ are prescribed, several of which were prescribed by [29], whose detailed presentation is deferred to Appendix N for space reasons. We note that the methods of [25] are designed for a different setting than that considered herein, namely that in which conformal coverage is sought adaptively over data streams.

In this vein, [40] have recently proposed a vector-score extension as that discussed herein, in which candidate weight vectors $\{w_m\} \in \mathbb{R}^K$ are searched over for score aggregation. That is, a vector $s(x) := (s_1(x, y), ..., s_K(x, y)) \in \mathbb{R}^K$ of scores $s_k(x, y)$ corresponding to each predictor $f_k(x)$ is predicted and its aggregate prediction region defined on the projection $\langle w_{m^*}, s \rangle$ for $w_{m^*}$ the weight resulting in the smallest prediction region. This method, however, has two shortcomings addressed herein. The first is that their method can only be applied in classification settings, whereas our method can be leveraged across both regression and classification problems. The second is that their approach only uses a *single* weighted projection in the end, resulting in suboptimal aggregation and, therefore, conservative prediction regions.

## 3 Method

### 3.1 Multivariate Score Quantile

We consider the setting typical of conformal model aggregation, as discussed in Section 2.4, in which predictors $f_1(x), ..., f_K(x)$ and corresponding scores $s_1(x, y), ..., s_K(x, y)$ are defined. We assume a similar premise as [40], in which the scores are stacked into a multivariate score $s(x, y) := (s_1(x, y), ..., s_K(x, y))$. A naive approach would then leverage standard conformal prediction over a pre-defined map $g : \mathbb{R}^K \to \mathbb{R}$, e.g., $g(s) = \sum_{k=1}^{K} s_k$. Similar to the naive conformalization of an ensembled predictor discussed in Section 2.4, using a *fixed* $g$ fails to adapt to any disparities in uncertainties present across predictors or requires intimate knowledge of such uncertainties. We instead wish to provide a data-adaptive pipeline to automatically produce such a $g$.

Importantly, we hereafter assume the score functions are non-negative, i.e., $s_k : \mathcal{X} \times \mathcal{Y} \to \mathbb{R}_+$, which is typically the case as the score serves as a generalization of the residual. We highlight that many of the details of the method presented below are geometric in nature and are more easily understood with the supplement of diagrams. We have, thus, provided an accompanying visual walkthrough of the procedure in Appendix A to clarify its presentation.

#### 3.1.1 Score Partial Ordering

Intuitively, our method seeks to directly generalize the approach of split conformal, by "ordering" the collection of multivariate calibration scores and taking the $1 - \alpha$ score under such an ordering to be a threshold $\widehat{\mathcal{Q}}$ with which prediction regions are then implicitly defined. Formally, the multivariate "ordering" is established as a pre-ordering $\lesssim$ over $\mathbb{R}^K$; a pre-ordering differs from a total ordering in that it need not satisfy the antisymmetric axiom of a total ordering. Roughly speaking, an "acceptance region," so called as it serves as the criterion used to ultimately decide which $y$ are accepted into the prediction region, is then defined as $\widehat{\mathcal{Q}} := \{s \mid s \lesssim \widehat{q}\}$, where $\widehat{q}$ is the $1 - \alpha$ empirical quantile of $\mathcal{S}_C$ under $\lesssim$. Such a $\widehat{\mathcal{Q}}$ naturally generalizes the standard scalar acceptance interval of $[0, \widehat{q}]$ in the case of non-negative score functions. We briefly highlight the distinction between *acceptance regions* and *prediction regions*. The former are subsets of the space $\subset \mathbb{R}^K$ of multivariate scores that ultimately define the criteria for retaining particular $y$ values in the prediction region. The latter are the subsets of the output space $\mathcal{Y}$ and it is these that ultimately have coverage guarantees. The two, however, are directly related; in particular, for a fixed score $s(x, y)$, a larger acceptance region will result in a more conservative prediction region.

Crucially, therefore, the problem of choosing this pre-ordering closely parallels that of choosing $g$, where a poorly chosen pre-ordering will result in overly large acceptance regions and, hence, conservative prediction regions. For instance, using a lexicographical ordering $\lesssim_{\text{Lex}}$ will result in axis-aligned hyper-rectangular acceptance regions. As a result, rather than manually prescribing a pre-ordering, we define $\lesssim$ in a data-driven fashion by prescribing an indexed family of nested sets $\{\mathcal{A}_t\}_{t \in \mathbb{R}}$, such that $\mathcal{A}_{t_1} \subset \mathcal{A}_{t_2}$ for $t_1 \leq t_2$ and stating $s_1 \lesssim s_2$ if $\forall t, s_2 \in \mathcal{A}_t \implies s_1 \in \mathcal{A}_t$.

For a family of sets $\{\mathcal{A}_t\}_{t \in \mathbb{R}}$, we take each $\mathcal{A}_t$ to be the region of the positive orthant $\mathbb{R}_+^K$ bounded by the coordinate axes and an "outer frontier" parameterized by $t$. The shape of this outer frontier remains fixed over the family and is merely scaled outward from the origin with $t$. Under this choice, comparing $s_1, s_2 \in \mathbb{R}^K$, i.e., checking if $s_1 \lesssim s_2$, amounts to checking if $t(s_1) \leq t(s_2)$, where $t(s)$ is the smallest $t$ for which the outer frontier of $\mathcal{A}_t$ intersects $s$. Notably, $t(s)$ is precisely the aforementioned data-driven score fusion function $g(s)$ of interest. Defining a data-adaptive $g(s)$, therefore, reduces to having a data-driven approach for defining the outer frontier of $\mathcal{A}_t$. We restrict this outer frontier to be such that $\mathcal{A}_t$ is a convex set; if $\mathcal{A}_t$ were permitted to be nonconvex, computing $t(s) := \min\{t \in \mathbb{R} : s \in \mathcal{A}_t\}$ would potentially be computationally expensive. The benefits of such convexity are highlighted, for example, in Section 3.2.

To have tight acceptance regions, we formally wish for the pre-ordering to have the property that the acceptance region given by $\widetilde{\mathcal{Q}}$ has minimal Lebesgue measure and captures $1 - \alpha$ points of $\mathcal{S}_C$. The problem of discovering an optimal pre-ordering can, thus, be equivalently stated as seeking to define the outer frontier of $\mathcal{A}_t$ to match that of the tightest $1 - \alpha$ convex cover of $\mathcal{S}_C$.

This motivates selecting the outer frontier to be the $1 - \alpha$ quantile envelope of $\mathcal{S}_C$. Using $\mathcal{S}_C$ to define $\mathcal{A}_t$ and in turn $\lesssim$, however, sacrifices the exchangeability of its points with test scores $s'$, as the very nature of ordering would change in swapping $s'$ with any $s \in \mathcal{S}_C$. The goal follows as seeking to define the outer frontier as the $1 - \alpha$ quantile envelope of $\mathcal{S}_C$ without directly using $\mathcal{S}_C$. For this reason, we partition $\mathcal{S}_C = \mathcal{S}_C^{(1)} \cup \mathcal{S}_C^{(2)}$, where we define $\lesssim$ using $\mathcal{S}_C^{(1)}$ and compute $\widehat{q}$ over $\mathcal{S}_C^{(2)}$. Such a split is predicated on the assumption that the $1 - \alpha$ quantile envelope defined over $\mathcal{S}_C^{(1)}$ resembles that of $\mathcal{S}_C^{(2)}$, implying the $|\mathcal{S}_C^{(1)}|$ should be sufficiently large as to capture this structure accurately.

We now focus attention on defining the quantile envelope over $\mathcal{S}_C^{(1)}$ using a technique paralleling that described in Section 2.2. In particular, we start by selecting the projection directions $\{u_m\}$ of Equation (1); since $s \in \mathbb{R}_+^K$, we similarly restrict $u_m \in \mathcal{S}_+^{K-1} := \mathcal{S}^{K-1} \cap \mathbb{R}_+^K$. To best approximate Equation (1), we wish for $\{u_m\}$ to be uniformly distributed over $\mathcal{S}_+^{K-1}$; however, exactly finding an evenly distributed set of points over hyperspheres in arbitrary $n$-dimensional spaces is a classically difficult problem [41] If $K = 2$, we can solve this exactly; for $K > 2$, we generate directions stochastically such that $U \sim \text{Unif}(\mathcal{S}_+^{K-1})$ by drawing $V_1, ..., V_M \sim \mathcal{N}(0, I^{K \times K})$ and defining $U_i := V_i^{|\cdot|} / \sqrt{V_1^2 + ... + V_M^2}$, where $v^{|\cdot|}$ denotes the component-wise absolute values.

We now wish to define the quantile thresholds $\{\widetilde{q}_m\}$ for the selected directions to optimally capture $1 - \alpha$ of $\mathcal{S}_C^{(1)}$. Naively taking the $1 - \alpha$ quantile per projection direction $u_m$ results in *joint* coverage by $\widetilde{\mathcal{Q}} := \bigcap_{m=1}^M H(u_m, \widetilde{q}_m)$ of $\mathcal{S}_C^{(1)}$ to be $< 1 - \alpha$. A straightforward fix is to replace the $1 - \alpha$ quantile per direction instead with its Bonferroni-corrected $1 - \alpha/M$ quantile. While valid, this approach produces overly conservative prediction regions. We, therefore, instead tune a separate $\beta \in (\alpha/M, \alpha)$ parameter via binary search, finding the maximum $\beta^*$ such that using the $\beta^*$ quantile per direction provides the overall desired coverage, i.e., $|\bigcap_{m=1}^M H(u_m, \widetilde{q}_m(1-\beta^*)) \cap \mathcal{S}_C^{(1)}| / N_{\mathcal{C}_1} \in (1-\alpha, 1-\alpha+\epsilon)$ for some fixed, small $\epsilon > 0$. With this choice of $\{(u_m, \widetilde{q}_m)\}$, we have a defined pre-ordering, whose coverage guarantees are formally stated below and proven in Appendix B.

**Theorem 3.1.** *Suppose* $(X_1, Y_1), \ldots, (X_{N_C}, Y_{N_C}), (X', Y')$ *are exchangeable, where* $\mathcal{D}_C := \{(X_i, Y_i)\}_{i=1}^{N_C}$. *Assume further that* $K$ *non-negative maps* $s_k : \mathcal{X} \times \mathcal{Y} \to \mathbb{R}_+$ *have been defined and a composite* $s(X, Y) := (s_1(X, Y), ..., s_K(X, Y))$ *is defined.*

*Let* $\sigma = (\sigma_1, \ldots, \sigma_{N_C})$ *be a random permutation of the indices* $\{1, \ldots, N_C\}$, *drawn uniformly and independently of* $\mathcal{D}_C$ *and* $(X', Y')$. *Let the calibration set* $\mathcal{D}_C$ *be partitioned into* $\mathcal{D}_C^{(1)} := \{(X_{\sigma_j}, Y_{\sigma_j})\}_{j=1}^{N_{C_1}}$ *and* $\mathcal{D}_C^{(2)} := \{(X_{\sigma_j}, Y_{\sigma_j})\}_{j=N_{C_1}+1}^{N_{C_1}+N_{C_2}}$, *where* $N_C := N_{C_1} + N_{C_2}$. *Let the corre-*

sponding score sets be $\mathcal{S}_C^{(1)}$ and $\mathcal{S}_C^{(2)}$. Let $T(\cdot; \mathcal{S}_C^{(1)}) : \mathbb{R}_+^K \to \mathbb{R}$ be a deterministic function for any given realization of $\mathcal{S}_C^{(1)}$.

For some $\alpha \in (0, 1)$, let $\hat{t}$ be the $\lceil (N_{C_2} + 1)(1 - \alpha) \rceil$-th smallest value of the set of transformed scores $\{T(s_i; \mathcal{S}_C^{(1)}) \mid s_i \in \mathcal{S}_C^{(2)}\}$. Assume that ties among the transformed scores occur with probability zero. Then, denoting by $\mathcal{C}(X') = \{y \in \mathcal{Y} \mid T(s(X', y); \mathcal{S}_C^{(1)}) \leq \hat{t}\}$, $\mathcal{P}(Y' \in \mathcal{C}(X')) \geq 1 - \alpha$, where the probability is defined over the joint draw of the data $\mathcal{D}_C$, $(X', Y')$, and the permutation $\sigma$.

### 3.1.2  Score Quantile Threshold

To then compute $\widehat{q}$, we find $t^*(s)$ for each $s \in \mathcal{S}_C^{(2)}$, defined to be $\min\{t \in \mathbb{R} : s \in \bigcap_{m=1}^M H(u_m, t\widetilde{q}_m)\}$. This can be efficiently computed as $t^*(s) = \max_{m=1,\ldots,M}(u_m^\top s / \widetilde{q}_m)$. Denoting the $\lceil (N_{C_2} + 1)(1 - \alpha) \rceil$-th largest $t^*(s)$ as $\widehat{t}$, $\widehat{q}_m := \widehat{t}\widetilde{q}_m$ and $\widehat{\mathcal{Q}} := \bigcap_{m=1}^M H(u_m, \widehat{q}_m)$. If the tightest quantile envelope was already discovered over $\mathcal{S}_C^{(1)}$, this adjustment factor $\widehat{t} \approx 1$. Critically, such calculations can be computed efficiently in vector form. Due to space restrictions, we defer this discussion to Appendix E. We additionally there empirically validate the efficiency of the procedure under this vectorized implementation. We present the full algorithm in Algorithm 1.

Importantly, while this procedure will result in convex regions $\widehat{\mathcal{Q}}$, this does **not** mean the downstream prediction regions in $\mathcal{Y}$ will be convex, as discussed in Section 3.2. However, it is unsurprising such flexibility exists, as even a single *scalar* score $s_1(x, y)$ can produce nonconvex prediction regions. One additional notable property of the CSA prediction regions is that their sizes vary across $x$ even if such variability is not baked into the constituent scores. For instance, using $s_k(x, y) := |f_k(x) - y|$ with standard, scalar conformal prediction yields intervals of length $2\widehat{q}_k$ for *any* $x$, yet $|\mathcal{C}^{\text{CSA}}(x)|$ even with such $\{s_k(x, y)\}$ *will* vary with $x$. This variability is desirable, as predictive uncertainty is seldom uniform across the covariate space. See Appendix F for a full illustration of this.

---

**Algorithm 1** CSA: UNIFHYPERSPHERE$(K)$ is an assumed subroutine that samples $\sim \text{Unif}(\mathcal{S}^{K-1})$.

1: **Inputs:** Score functions $s_1, \ldots, s_K : \mathcal{X} \to \mathcal{Y}$, Calibration set $\mathcal{D}_C$, Desired coverage $1 - \alpha$
2: $[\beta_{\text{lo}}, \beta_{\text{hi}}] \leftarrow [\alpha/M, \alpha]$, $\widehat{\mathcal{Q}} \leftarrow \emptyset$
3: $\sigma \sim \text{Unif}(\text{Permutations of } \{1, \ldots, N_C\})$
4: $\mathcal{S}_C^{(1)} \cup \mathcal{S}_C^{(2)} \leftarrow \{(s_k(x_{\sigma(i)}, y_{\sigma(i)}))_{k=1}^K\}_{i=1, N_{C_1}+1}^{N_{C_1}, N_{C_2}}$, $\quad \{u_m \leftarrow \text{UNIFHYPERSPHERE}(K)\}_{m=1}^M$
5: **while** $|\mathcal{S}_C^{(1)} \bigcap \widehat{\mathcal{Q}}| / N_{C_1} \notin 1 - \alpha \pm \epsilon$ **do**
6: $\quad \beta \leftarrow (\beta_{\text{lo}} + \beta_{\text{hi}})/2$
7: $\quad \left\{\widetilde{q}_m \leftarrow (1 - \beta) \text{ empirical quantile of } \{u_m^\top s_i\}_{s_i \in S_C^{(1)}}\right\}_{m=1}^M$
8: $\quad \widehat{\mathcal{Q}} \leftarrow \bigcap_{m=1}^M H(u_m, \widetilde{q}_m)$
9: $\quad$ **if** $|\mathcal{S}_C^{(1)} \bigcap \widehat{\mathcal{Q}}| / N_{C_1} > 1 - \alpha$ **then** $\beta_{\text{lo}} \leftarrow \beta$ **else** $\beta_{\text{hi}} \leftarrow \beta$
10: **end while**
11: $\widehat{t} \leftarrow (1 - \alpha)$ empirical quantile of $\{\max_{m \in [M]}(u_m^\top s_i / \widetilde{q}_m)\}_{s_i \in S_C^{(2)}}$
12: **Return** $\{(u_m, \widehat{t}\widetilde{q}_m)\}_{m=1}^M$

---

Notably, this algorithm achieves the aforementioned coverage guarantee as a direct corollary of Theorem 3.1, stated below and proven in Appendix C. Intuitively, the proof proceeds by demonstrating that the $T$ scoring function defined implicitly by Algorithm 1 satisfies those conditions posited in Theorem 3.1, from which the posited coverage immediately follows.

**Corollary 3.2.** *Let $\mathcal{D}_C$, $(X', Y')$, $\{s_k\}_{k=1}^K$, and $\alpha$ be as defined in Theorem 3.1. Let $\sigma$, $(\mathcal{S}_C^{(1)}, \mathcal{S}_C^{(2)})$, and $U = \{u_m\}_{m=1}^M$ be as defined by lines 3-4 of the call $\text{CSA}(\{s_k\}, \mathcal{D}_C, 1 - \alpha)$ of Algorithm 1. Denote by $\{\widetilde{q}_m\}_{m=1}^M$ the parameters defined by lines 4-9 of Algorithm 1 and by $T$ the scoring function $T(s; \mathcal{S}_C^{(1)}, U) = \max_{m=1,\ldots,M}(u_m^\top s / \widetilde{q}_m)$ for any score vector $s \in \mathbb{R}_+^K$. Then, denoting by $\mathcal{C}(X') = \{y \in \mathcal{Y} \mid T(s(X', y); \mathcal{S}_C^{(1)}) \leq \hat{t}\}$, $\mathcal{P}(Y' \in \mathcal{C}(X')) \geq 1 - \alpha$, where the probability is defined over the joint draw of the data $\mathcal{D}_C$, $(X', Y')$, and the permutation $\sigma$.*

## 3.2 Predict-Then-Optimize

With this generalization of the score function, a natural question is how to leverage the resulting prediction regions $\mathcal{C}(x)$. For both classification and regression, $\mathcal{C}(x) = \bigcap_{m=1}^{M} \mathcal{C}_m(x)$ where $\mathcal{C}_m(x) := \{y \mid u_m^\top s(x, y) \leq \widehat{q}_m\}$. For *classification*, where $|\mathcal{Y}| \in \mathbb{N}$, explicit construction of $\mathcal{C}(x)$ is straightforward: for any $x$, explicitly constructing $\mathcal{C}(x)$ can be done by iterating through $y \in \mathcal{Y}$ and checking if $s(x, y) \in \widehat{\mathcal{Q}}$ by comparing $s(x, y)$ against each one of the thresholds $\widehat{q}_m$ after projection.

In the case of regression, however, the prediction region cannot be explicitly constructed in the general case, since $\mathcal{Y}$ contains uncountably many elements. In fact, explicit construction is generally not of interest for downstream regression applications. We, therefore, focus on one particular application, namely that of [36] discussed in Section 2.3, and demonstrate the CSA prediction regions can be leveraged in their framework for problems studied therein. For instance, the authors demonstrated the utility of their method in a robust traffic routing setting with $c$ being predicted traffic from a probabilistic weather model $q(C \mid X)$ for weather covariates $X$. An ensembling approach emerges with multiple predictive models, such as a $q_2(C \mid X)$ predicting traffic based on historical trends.

As in Section 3.1, we note that the below described algorithm is better understood with a visual accompaniment, which we provide in Appendix D. [36] demonstrated that solving the robust problem variant $w^*(x) := \min_w \max_{\widehat{c} \in \mathcal{C}(x)} f(w, \widehat{c})$ in a computationally efficient manner is feasible by performing gradient-based optimization on $w$, where the gradient $\nabla_w \phi(w)$ of $\phi(w) := \max_{\widehat{c} \in \mathcal{C}(x)} f(w, \widehat{c})$ can be computed by leveraging Danskin's Theorem so long as $\max_{\widehat{c} \in \mathcal{C}(x)} f(w, \widehat{c})$ is efficiently computable for any fixed $w$. We focus on demonstrating that this remains the case for CSA, specifically considering the case where individual view score functions take the form of the "GPCP" score considered therein. In this setup, each constituent predictor is a generative model $q_k(C \mid X)$ from which $\{\widehat{c}_{kj}\}_{j=1}^{J_k} \sim q_k(C \mid X)$ samples are drawn. Note that $J_k$ need not be constant across $k$. The GPCP score, used to define the score components, is

$$s_k(x, c) = \min_{j \in 1, \ldots, J_k} \left[ ||\widehat{c}_{kj} - c||_2 \right]. \tag{2}$$

Notably, this framework subsumes many standard regression settings, e.g., for a deterministic predictor, one can take $q_k(C \mid X) = \delta(f_k(X))$. To compute $\max_{\widehat{c} \in \mathcal{C}(x)} f(w, \widehat{c})$, we first let $\vec{j} \in \mathcal{J} = \{j_1, \ldots, j_K\}$ be an indexing tuple, where each $j_k \in \{1, \ldots, J_k\}$. That is, each $\vec{j}$ is a vector that "selects" one sample per predictor. Notably then, the projection $u_m^\top s(\widehat{c}_{\vec{j}}, c)$ is convex in $c$, since the projection directions are all restricted to $\mathcal{S}_+^{K-1}$. Thus,

$$c_{\vec{j}}^* := \arg\max_c f(w, c) \qquad \text{s.t.} \qquad u_m^\top s(\widehat{c}_{\vec{j}}, c) \leq \widehat{q}_m \quad \forall m \in \{1, \ldots, M\} \tag{3}$$

remains a standard convex optimization problem. The final maximum can then be found by aggregation, namely $c^* = \arg\max_{\vec{j} \in \mathcal{J}} f(w, c_{\vec{j}}^*)$. While $|\mathcal{J}| = \prod_{k=1}^{K} J_k$, in certain cases of ensemble prediction, such as multi-view prediction, there tend to be a limited number of predictors in practice, typically $K = 2$ or $K = 3$. This coupled with the trivial parallelizability of computing over indices means this approach is still computationally tractable. The full procedure is outlined in Algorithm 2.

---

**Algorithm 2** Predict-Then-Optimize Under CSA

---

1: **Inputs:** Context $x$, Predictors $\{q_k(C \mid X)\}_{k=1}^{K}$, Optimization steps $T$, Sample counts $\{J_k\}_{k=1}^{K}$, CSA quantile $\{(u_m, \widehat{q}_m)\}_{m=1}^{M}$
2: $\{\{\widehat{c}_{kj}\}_{j=1}^{J_k} \sim q_k(C \mid X)\}_{k=1}^{K}$, $\mathcal{J} = \prod_{k=1}^{K}[J_k]$
3: $w^{(0)} \sim U(\mathcal{W})$
4: **for** $t \in \{1, \ldots T\}$ **do**
5: $\quad$ **for** $\vec{j} \in \mathcal{J}$ **do** $c_{\vec{j}}^* \leftarrow \arg\max_c f(w^{(t)}, c)$ $\qquad$ s.t. $\quad \forall m \in 1, \ldots, M$ $\qquad u_m^\top s(\widehat{c}_{\vec{j}}, c) \leq \widehat{q}_m$
6: $\quad$ $c^* \leftarrow \arg\max_{c_{\vec{j}}^*} f(w^{(t)}, c_{\vec{j}}^*)$
7: $\quad$ $w^{(t)} \leftarrow \Pi_{\mathcal{W}}(w^{(t-1)} - \eta \nabla_w f(w^{(t-1)}, c^*))$
8: **end for**
9: **Return** $w^{(T)}$

---

Table 1: Classification results are shown across tasks for $\alpha = 0.10$, $\alpha = 0.05$, and $\alpha = 0.01$, with coverages in the top (grey) and average prediction set sizes (white) in the bottom of each row. Both were assessed over a batch of i.i.d. test samples (15% of the validation set from ImageNet). Standard deviations and means were computed across 10 randomized draws of the calibration and test sets.

| Dataset/$\alpha$ | ResNet | VGG | DenseNet | VFCP | $\mathcal{C}^M$ | $\mathcal{C}^R$ | $\mathcal{C}^U$ | Ensemble | CSA |
|---|---|---|---|---|---|---|---|---|---|
| ImageNet | 0.901 (0.005) | 0.902 (0.003) | 0.902 (0.003) | 0.899 (0.004) | 0.938 (0.003) | 0.909 (0.004) | 0.9 (0.004) | 0.899 (0.004) | 0.9 (0.003) |
| ($\alpha = 0.10$) | 137.004 (1.98) | 136.116 (2.206) | 120.096 (2.427) | 46.063 (1.089) | 87.337 (1.604) | 82.746 (1.692) | 131.856 (2.378) | 69.123 (1.317) | **34.006 (0.924)** |
| ($\alpha = 0.05$) | 0.95 (0.003) | 0.949 (0.004) | 0.952 (0.002) | 0.95 (0.003) | 0.975 (0.002) | 0.954 (0.004) | 0.95 (0.003) | 0.949 (0.002) | 0.95 (0.003) |
| | 220.022 (2.072) | 229.523 (3.076) | 208.658 (2.016) | 78.108 (2.004) | 166.933 (2.157) | 143.323 (2.932) | 220.491 (2.773) | 112.161 (2.115) | **59.574 (3.382)** |
| ($\alpha = 0.01$) | 0.99 (0.001) | 0.991 (0.001) | 0.989 (0.002) | 0.99 (0.001) | 0.997 (0.001) | 0.991 (0.002) | 0.99 (0.002) | 0.99 (0.002) | 0.99 (0.002) |
| | 491.952 (6.353) | 726.028 (12.157) | 459.399 (6.739) | **194.691 (4.579)** | 580.592 (7.715) | 532.155 (24.829) | 559.188 (7.07) | 299.453 (6.526) | **201.32 (46.509)** |

## 4 Experiments

We now study CSA empirically across several tasks, demonstrating its coverage guarantees with reduced conservatism. We demonstrate improvements in an ImageNet classification task in Section 4.1, across real-data regression benchmark tasks in Section 4.2 as proposed by [42], and in a downstream predict-then-optimize task in Section 4.3. We additionally assess the robustness of CSA to imbalanced ensembles and perform an ablation study of the two-stage calibration.

We note that the predictors and calibration and test sets were fixed across choices of calibration procedure for each experiment, meaning care had to be taken in partitioning $\mathcal{D}_\mathcal{C} = \mathcal{D}_\mathcal{C}^{(1)} \cup \mathcal{D}_\mathcal{C}^{(2)}$ for CSA, where an insufficiently large $\mathcal{D}_\mathcal{C}^{(1)}$ would result in poor estimation of the $\alpha$-quantile envelope and hence require a large adjustment $\widehat{t}$ factor and an insufficiently large $\mathcal{D}_\mathcal{C}^{(2)}$ in the classical reduced predictive efficiency from conformal prediction. We note the splits in each of the sections that follow.

We compare against the methods presented in Section 2.4, viz. the model selection of [27], the aggregation methods of [25], and the single weighted score projection (VFCP) of [40]. We additionally include the initial strategy discussed in Section 2.4, in which the ensemble predictor is directly conformalized, using a natural aggregate "ensemble" score, given in the following sections. From the work of [25], we consider the following methods: the standard majority-vote $\mathcal{C}^M$, partially randomized thresholding $\mathcal{C}^R$, and fully randomized thresholding $\mathcal{C}^U$ approaches (see Appendix N). Notably, these methods do not lend themselves for use in the predict-then-optimize setting, so we eliminate them from consideration therein. VFCP can only be applied in classification settings; we, thus, do not compare to it across the regression tasks. Code is available at https://github.com/yashpatel5400/fusioncp/.

### 4.1 Classification Tasks

We first study the predictive efficiency of the aforementioned methods on the ImageNet classification task [43]. In particular, an ensemble was constructed from three separately trained deep learning architectures, namely ResNet-50, VGG-11, and DenseNet-121. Conformalization on the individual models was performed using the standard classification score function across all approaches, namely $s(x, y) = \sum_{j=1}^{l} \widehat{f}(x)_{\pi_j(x)}$ where $y = \pi_l(x)$ and $\pi(x)$ is the permutation of $\{1, \ldots, |\mathcal{Y}|\}$ that sorts $\widehat{f}(x)$ from most to least likely. Here, the "Ensemble" score was computed with the same $s(x, y)$, replacing $f_k(x)$ with the ensemble average probability, i.e. $\mu(x)_j := \sum_k (f_k(x))_j / K$. Calibration was performed using 85% of the ImageNet test set and assessment of the coverage and interval lengths on the remaining 15%, with 10 trials conducted over randomized draws of these calibration and test sets. A 25/75% split was used for $\mathcal{D}_\mathcal{C}^{(1)}$-$\mathcal{D}_\mathcal{C}^{(2)}$. The results are presented in Table 1; the full results across additional $\alpha$ is given in Appendix G. We see that all the approaches exhibit the desired coverages across $\alpha$. However, CSA consistently produces significantly smaller prediction regions than both the individually conformalized models and alternate aggregation strategies.

### 4.2 Regression Tasks

We now similarly study the predictive efficiency of CSA across a suite of regression tasks from [42]. The data for each task were split with 50/45/5% for training, calibration, and testing for coverage and interval lengths, with five trials conducted over randomized selections of such sets. A 5/95% split was used for $\mathcal{D}_\mathcal{C}^{(1)}$-$\mathcal{D}_\mathcal{C}^{(2)}$. The problem setup was replicated from [29], in which four prediction methods were ensembled, namely an OLS model, a LASSO linear model, a random forest (RF),

Table 2: The results for five distinct tasks are shown below for $\alpha = 0.05$ (top five rows) and $\alpha = 0.025$ (bottom five rows). For each, the average coverages (grey rows) and prediction set lengths (white rows) with standard deviations are given, both assessed over 5 randomized draws of the training, calibration, and test sets. In cases where the method failed to achieve sufficient coverage (i.e. $< .93$ for $\alpha = 0.05$ and $< 0.96$ for $\alpha = 0.025$), we do not include it in comparison for set length.

| Dataset/$\alpha$ | OLS | LASSO | RF | XGBoost | $\mathcal{C}^M$ | $\mathcal{C}^R$ | $\mathcal{C}^U$ | Ensemble | Single-Stage | CSA |
|---|---|---|---|---|---|---|---|---|---|---|
| 361234 ($\alpha=0.05$) | 0.97 (0.011) | 0.966 (0.011) | 0.939 (0.002) | 0.954 (0.006) | 0.956 (0.011) | 0.948 (0.01) | 0.96 (0.013) | 0.95 (0.006) | 0.955 (0.013) | 0.957 (0.01) |
|  | 9.673 (0.160) | 9.645 (0.154) | 10.080 (0.160) | 9.157 (0.052) | 9.196 (0.123) | 8.703 (0.086) | 9.524 (0.056) | 17.759 (0.275) | 7.646 (0.073) | **7.688 (0.181)** |
| 361235 ($\alpha=0.05$) | 0.947 (0.0) | 0.945 (0.005) | 0.968 (0.016) | 0.95 (0.005) | 0.955 (0.016) | 0.897 (0.005) | 0.953 (0.011) | 0.932 (0.021) | 0.745 (0.011) | 0.984 (0.005) |
|  | 20.961 (0.651) | 24.241 (0.246) | **10.096 (0.587)** | 11.387 (0.452) | 11.782 (0.057) | — | 16.088 (0.118) | 15.823 (1.272) | 6.162 (0.458) | 11.695 (0.266) |
| 361236 ($\alpha=0.05$) | 0.975 (0.008) | 0.975 (0.008) | 0.961 (0.0) | 0.948 (0.012) | 0.948 (0.012) | 0.938 (0.012) | 0.965 (0.008) | 0.934 (0.004) | 0.94 (0.004) | 0.963 (0.004) |
|  | 4.44e4 (1.17e3) | 4.45e4 (1.23e3) | 5.08e4 (3.86e2) | 4.10e4 (1.22e3) | 4.32e4 (1.00e3) | 4.09e4 (1.09e3) | 4.44e4 (8.52e2) | 6.05e4 (2.41e3) | 3.10e4 (2.48e3) | **3.34e4 (1.28e3)** |
| 361237 ($\alpha=0.05$) | 0.969 (0.023) | 0.969 (0.023) | 0.981 (0.0) | 0.923 (0.0) | 0.954 (0.015) | 0.9 (0.008) | 0.969 (0.023) | 0.885 (0.038) | 0.8 (0.015) | 0.977 (0.008) |
|  | 44.019 (0.990) | 44.069 (1.115) | 27.035 (1.014) | — | 26.524 (1.244) | — | 31.967 (1.118) | — | 14.473 (0.503) | **23.145 (0.199)** |
| 361241 ($\alpha=0.05$) | 0.954 (0.001) | 0.956 (0.001) | 0.944 (0.005) | 0.957 (0.002) | 0.954 (0.002) | 0.923 (0.0) | 0.952 (0.0) | 0.949 (0.001) | 0.917 (0.006) | 0.951 (0.001) |
|  | 19.133 (0.062) | 20.245 (0.095) | 18.102 (0.055) | 18.482 (0.054) | 17.958 (0.062) | — | 18.932 (0.034) | 29.548 (0.191) | 15.199 (0.427) | **17.328 (0.097)** |
| 361234 ($\alpha=0.025$) | 0.987 (0.008) | 0.987 (0.008) | 0.974 (0.004) | 0.977 (0.008) | 0.982 (0.008) | 0.971 (0.01) | 0.981 (0.01) | 0.97 (0.008) | 0.976 (0.01) | 0.973 (0.006) |
|  | 11.939 (0.137) | 11.871 (0.084) | 12.484 (0.168) | 11.972 (0.009) | 11.587 (0.110) | 11.157 (0.086) | 11.965 (0.050) | 25.598 (0.974) | 9.306 (0.259) | **8.855 (0.059)** |
| 361235 ($\alpha=0.025$) | 0.987 (0.0) | 0.982 (0.011) | 0.979 (0.011) | 0.984 (0.005) | 0.989 (0.005) | 0.966 (0.016) | 0.976 (0.005) | 0.958 (0.021) | 0.889 (0.011) | 0.989 (0.005) |
|  | 24.595 (0.825) | 28.841 (1.129) | **11.811 (0.992)** | 14.237 (0.786) | 14.472 (0.172) | 12.278 (0.026) | 19.231 (0.356) | — | 7.719 (0.467) | 12.563 (0.766) |
| 361236 ($\alpha=0.025$) | 0.992 (0.004) | 0.992 (0.004) | 0.981 (0.0) | 0.965 (0.008) | 0.975 (0.008) | 0.965 (0.008) | 0.977 (0.012) | 0.955 (0.008) | 0.955 (0.012) | 0.973 (0.004) |
|  | 4.86e4 (8.74e2) | 4.86e4 (8.68e2) | 5.61e4 (3.69e2) | 4.66e4 (1.57e3) | 4.76e4 (8.10e2) | 4.57e4 (1.06e3) | 4.92e4 (7.53e2) | — | 3.29e4 (3.11e3) | **3.58e4 (1.95e3)** |
| 361237 ($\alpha=0.025$) | 0.981 (0.0) | 0.981 (0.0) | 0.981 (0.0) | 0.977 (0.008) | 0.962 (0.0) | 0.962 (0.0) | 0.977 (0.008) | 0.965 (0.008) | 0.927 (0.031) | 0.981 (0.0) |
|  | 47.738 (0.542) | 47.440 (0.959) | 30.785 (0.037) | **26.208 (0.897)** | 30.554 (0.561) | 27.182 (0.803) | 35.982 (0.619) | 67.660 (6.380) | 18.214 (0.436) | 26.897 (0.515) |
| 361241 ($\alpha=0.025$) | 0.979 (0.001) | 0.978 (0.001) | 0.976 (0.001) | 0.978 (0.001) | 0.978 (0.0) | 0.964 (0.002) | 0.977 (0.0) | 0.972 (0.002) | 0.958 (0.003) | 0.979 (0.0) |
|  | 21.772 (0.085) | 23.089 (0.106) | 21.543 (0.009) | 21.454 (0.109) | 20.862 (0.088) | **19.291 (0.060)** | 21.905 (0.041) | 40.082 (0.045) | 17.765 (0.329) | 19.897 (0.062) |

and an XGBoost model. A residual function was used as the score across all methods, namely $s(x, y) = |\widehat{f}(x) - y|$. Here, the "Ensemble" score was the standard $s(x, y) := \frac{|\mu(x) - y|}{\sigma(x)}$, where $(\mu(x), \sigma(x))$ are the ensemble mean and standard deviation. Prediction intervals could be analytically constructed for the $\mathcal{C}^M$, $\mathcal{C}^R$, and $\mathcal{C}^U$ methods. To assess CSA, however, a discretized grid $\mathcal{G}_Y \subset \mathcal{Y}$ of coarseness $\Delta y$ was considered, and an interval length estimate given by $\mathcal{L}(\mathcal{C}(x)) \approx \Delta y \cdot |\{y : y \in \mathcal{G}_Y, s(x, y) \in \widehat{\mathcal{Q}}\}|$. We also present an ablation, labeled "Single-Stage," to demonstrate the two-stage calibration is necessary to retain coverage; this single-stage approach does not split $\mathcal{S}_C$ and instead directly computes $\{\widehat{q}_m\}$ on $\mathcal{S}_C$ per Section 3.1.1. For CSA, $M = 1000$ was used. Intuitively, as $M \to \infty$, we would expect to recover the $1 - \alpha$ tightest cover and, thus, that the prediction region size should be roughly decreasing in $M$, with some plateau. This is explicitly shown in Appendix I.

We provide the results for $\alpha = 0.05$ and $\alpha = 0.025$ to demonstrate the consistency of the method performance. A subset of the results is given in Table 2; the full set of results is deferred to Appendix H. As in the results of Section 4.1, we see that CSA retains the coverage guarantees typical of conformal prediction yet produces significantly smaller prediction intervals than both the individual models and the alternate aggregation strategies. We additionally see that the "Single-Stage" approach fails to retain coverage, demonstrating the necessity of the two-stage calibration. We provide a visual comparison of the prediction regions resulting from these methods in Appendix J.

We additionally assessed the robustness of our method to imbalanced ensembles. The experiments of [29] were conducted on a UCI benchmark task [44] with an ensemble of an OLS model, a LASSO linear model, a random forest, and an MLP, and they found the conformalized random forest to outperform all the proposed aggregation strategies, due to the lack of orthogonal information in considering the other predictors. We find that, in these degenerate cases, where the best decision is to simply choose a single predictor, our method outperforms other aggregation methods and nearly matches the performance of the best conformalized predictor in hindsight; the results are presented in Appendix K across a number of UCI benchmarks.

## 4.3 CSA Predict-Then-Optimize

We now study a real-world predict-then-optimize traffic routing task, from [36]. In this task, a time series of $T$ preceding precipitations is used to predict future precipitations and, in turn, future traffic, as fully described in Appendix L. We consider the traffic routing problem for a fixed source-target pair $(s, t)$ over the graph of Manhattan, where $|\mathcal{V}| = 4584$ and $|\mathcal{E}| = 9867$. Formally,

$$w^*(x) := \min_w \max_{\widehat{c} \in \mathcal{C}(x)} \widehat{c}^T w \quad \text{s.t.} \quad w \in [0, 1]^{\mathcal{E}}, Aw = b, \mathcal{P}_{X,C}(C \in \mathcal{C}(X)) \geq 1 - \alpha$$

where $x \in \mathbb{R}^{T \times H \times W}$ are the previous precipitation readings, $w_e \in \mathbb{R}^{|\mathcal{E}|}$ the traffic proportion routed along road $e$, $c \in \mathbb{R}^{|\mathcal{E}|}$ the transit times anticipated across roads, $A \in \mathbb{R}^{|\mathcal{V}| \times |\mathcal{E}|}$ the graph incidence

matrix, and $b \in \mathbb{R}^{|\mathcal{V}|}$ the vector that specifies the routing problem, in which $b_s = 1, b_t = -1$, and $b_k = 0$ for $s$ the travel source node, $t$ the terminal node, and $k \notin \{s, t\}$ all other nodes.

We then consider two probabilistic models for traffic prediction, namely one based on the classical probabilistic Lagrangian integro-difference approach (STEPS) of [45] and one on the modern latent diffusion model (LDM) approach of [46]. As a result of the higher inference cost of the latter, we consider the setup where $J_1 > J_2$, specifically with $J_1 = 4$ and $J_2 = 1$, highlighting the flexibility of non-uniform sampling from predictors discussed in Section 3.2. As discussed in Section 4, the alternate aggregation strategies do not lend themselves for use in this setting. We, therefore, only compare CSA to the separate conformalizations of the two predictors, with the score from Equation (2). We here evaluate the methods using the expected suboptimality gap proportion, $\Delta_\% = \mathbb{E}_X[\Delta(X, C(X))/\min_w f(w, C(X))]$, where $\Delta$ is defined as discussed in Section 2.3. This measures the conservatism of the robust optimal value and is bounded in $[0, 1]$.

Experiments were conducted with $|\mathcal{D}_C| = 200$, with a 20/80% split used for $\mathcal{D}_C{}^{(1)}$-$\mathcal{D}_C{}^{(2)}$. The suboptimality was then computed across 100 i.i.d. test samples. To assess the improvement, we conducted two paired t-tests, where $H_0 : \Delta_\%^{(\text{CSA})} = \Delta_\%^{(\text{STEPS})}$ and $H_1 : \Delta_\%^{(\text{CSA})} < \Delta_\%^{(\text{STEPS})}$ and similarly for $\Delta_\%^{(\text{CSA})}$ and $\Delta_\%^{(\text{LDM})}$. The results are provided in Table 3, from which we find that CSA significantly reduces the suboptimality after accounting for Bonferroni multiple testing. We see that, while conformalization of either of the two views individually already produces the desired coverage, CSA produces more informative prediction regions, and hence less conservative robust upper bounds.

Table 3: Coverages for $\alpha = 0.05$ for the individually conformalized and CSA approach and p-values of the paired t-tests comparing $\Delta_\%$ are shown, both computed over 100 i.i.d. test samples.

| Coverage | | | P-values for $H_1$ |
|---|---|---|---|
| STEPS | LDM | CSA | $\Delta_\%^{(\text{CSA})} < \Delta_\%^{(\text{STEPS})}: 3.61 \times 10^{-4}$ |
| 0.981 | 0.962 | 0.968 | $\Delta_\%^{(\text{CSA})} < \Delta_\%^{(\text{LDM})}: 9.50 \times 10^{-4}$ |

## 5 Discussion

We have presented a framework for producing informative prediction regions in ensemble predictor pipelines, suggesting many directions for extension. One is in extracting insights of the relative predictor uncertainties from the data-driven relation $\lesssim$. Another is the integration of CSA with [47], which proposed an end-to-end extension to [36]. Such end-to-end integration may discover more optimal vector comparisons than the quantile envelope partial ordering approach proposed herein. Additionally, in requiring the data to be split to retain coverage, we are sacrificing statistical efficiency, which may be infeasible in data-sparse regimes: developing a method analogous to full conformal, which forgoes computational efficiency for statistical efficiency, would be another valuable direction for extension. Finally, given the prevalence of sensor fusion in robotics, another avenue is to study the use of CSA in robust control. This would extend recent works that have leveraged conformal prediction for robust linear control [48].

**Acknowledgements**

We acknowledge the support of the College of LSA at the University of Michigan via the "Meet the Moment" initiative.

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

# A CSA Visual Walkthrough

We walk through a visual presentation of the approach below to supplement the textual description in the main text. We start with a collection of multivariate calibration scores $\mathcal{S}_C$ with $s \in \mathcal{S}$ being $\in \mathbb{R}^K$. For the purposes of visualization in this section, we have $K = 2$. We first partition the score evaluations $\mathcal{S}_\mathcal{C} = \mathcal{S}_C^{(1)} \cup \mathcal{S}_C^{(2)}$, with a subset $\mathcal{S}_C^{(1)}$ used to define the pre-ordering and the remainder $\mathcal{S}_C^{(2)}$ to define the multivariate quantile.

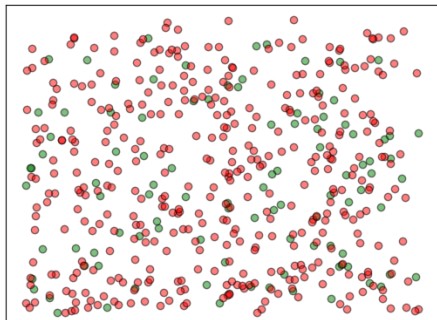

Figure 2: The calibration score evaluations are first split between those used to define the pre-ordering (green) $\mathcal{S}_C^{(1)}$ and those used to define the final multivariate quantile (red) $\mathcal{S}_C^{(2)}$.

We first wish to define the pre-ordering over $\mathcal{S}_C^{(1)}$. As described in the main text, the goal is to define this using an indexed family of sets $\mathcal{A}_t$ with index $t \in \mathbb{R}$, after which the multivariate quantile approach reduces to the univariate quantile formulation. To ensure the final envelope over $\mathcal{S}_C^{(2)}$ remains as tight as possible, we wish to define this family in a data-driven fashion. Critically, the *shape* of this tightest envelope around $\mathcal{S}_C^{(2)}$ will vary across $\alpha$, meaning we must define the family *separately* for each choice of $\alpha$. We expect the contour of the tightest $\alpha$ envelope for $\mathcal{S}_C^{(1)}$ will be similar to that over $\mathcal{S}_C^{(2)}$, motivating such a choice to define the indexing family. To do this, we project $\mathcal{S}_C^{(1)}$ along a number of directions, finding the $\beta$ quantile along each, in turn defining a half-plane, where $\beta$ is as described in Section 3.1.

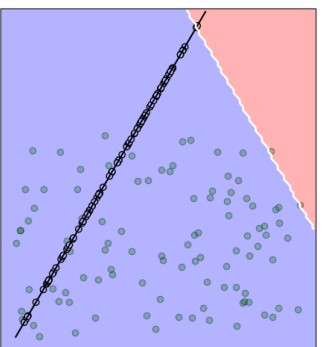 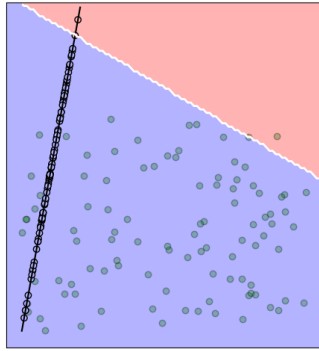

Figure 3: The pre-ordering points are projected across a number of directions, after which the $\beta$ quantile is used to define a direction quantile. This defines a half-plane of points that are in the region (blue) and those outside (red).

We then iteratively update $\beta$ in the manner described in Algorithm 1 to obtain $\beta^*$, namely the minimum value for which the region given by the intersection of the corresponding half-planes covers roughly $1 - \alpha$ of $\mathcal{S}_C^{(1)}$.

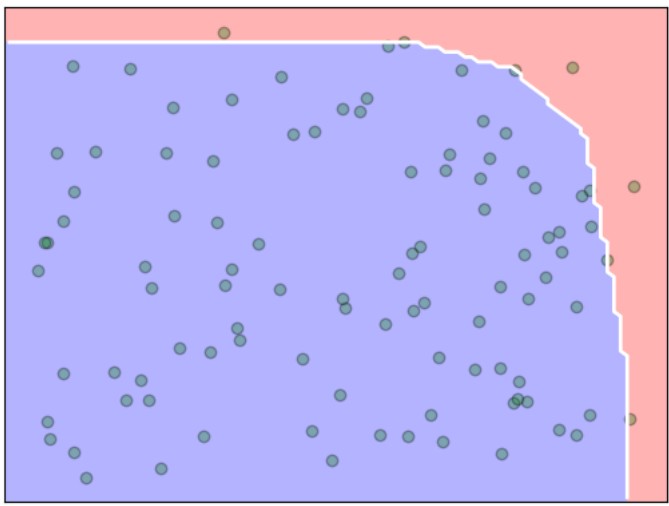

Figure 4: We use the intersection of hyperplanes to define the quantile envelope, seeking $\beta^*$ that achieves the desired coverage.

Once this $1 - \alpha$ quantile envelope of $\mathcal{S}_C^{(1)}$ is found, we define $\mathcal{A}_1$ to be such an envelope, with which future points can now be partially ordered. That is, for any point $s \in \mathbb{R}^K$ notice that we can unambiguously associate it with $t(s) := \min\{t \in \mathbb{R} : s \in \mathcal{A}_t\}$. Intuitively, this is the $t$ where the contour "intersects" $s$. Notably, now that the partial ordering has been defined, the points of $\mathcal{S}_C^{(1)}$ are no longer used. It would be of interest to investigate whether a concurrent definition of the partial ordering and final calibration is possible without such data splitting in future work.

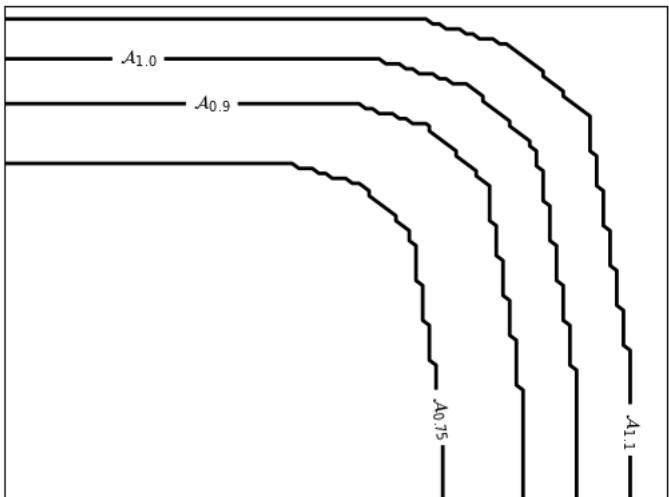

Figure 5: Using the quantile envelope, the family of nested sets $\mathcal{A}_t$ is defined, in turn defining a partial ordering over $\mathbb{R}^K$.

With this $\mathcal{A}_t$, we find the final $\widehat{q}$ simply by mapping the points of $\mathcal{S}_C^{(2)}$ to their corresponding $t(s)$ values in the aforementioned fashion and performing standard conformal prediction. As discussed, if the envelope has a similar structure to that found over $\mathcal{S}_C^{(1)}$, the envelope should be adjusted by only a minor amount.

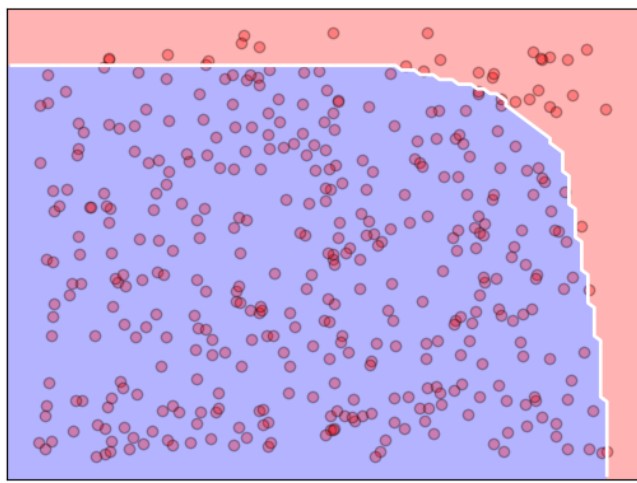

Figure 6: Using the nested family of sets, we expand or contract the envelope appropriately using the data of $\mathcal{S}_C^{(2)}$ to find the final adjustment factor.

# B  Multivariate Score Coverage

The proof of the multivariate extension of conformal prediction follows in precisely the same manner as that of standard conformal prediction with the pre-order $\precsim$ replacing the complete ordering used in traditional conformal prediction.

**Theorem B.1.** *Suppose* $(X_1, Y_1), \ldots, (X_{N_C}, Y_{N_C}), (X', Y')$ *are exchangeable, where* $\mathcal{D}_C := \{(X_i, Y_i)\}_{i=1}^{N_C}$. *Assume further that $K$ non-negative maps $s_k : \mathcal{X} \times \mathcal{Y} \to \mathbb{R}_+$ have been defined and a composite $s(X, Y) := (s_1(X, Y), ..., s_K(X, Y))$ is defined.*

*Let $\sigma = (\sigma_1, \ldots, \sigma_{N_C})$ be a random permutation of the indices $\{1, \ldots, N_C\}$, drawn uniformly and independently of $\mathcal{D}_C$ and $(X', Y')$. Let the calibration set $\mathcal{D}_C$ be partitioned into $\mathcal{D}_C^{(1)} := \{(X_{\sigma_j}, Y_{\sigma_j})\}_{j=1}^{N_{C_1}}$ and $\mathcal{D}_C^{(2)} := \{(X_{\sigma_j}, Y_{\sigma_j})\}_{j=N_{C_1}+1}^{N_{C_1}+N_{C_2}}$, where $N_C := N_{C_1} + N_{C_2}$. Let the corresponding score sets be $\mathcal{S}_C^{(1)}$ and $\mathcal{S}_C^{(2)}$. Let $T(\cdot; \mathcal{S}_C^{(1)}) : \mathbb{R}_+^K \to \mathbb{R}$ be a deterministic function for any given realization of $\mathcal{S}_C^{(1)}$.*

*For some $\alpha \in (0, 1)$, let $\hat{t}$ be the $\lceil (N_{C_2}+1)(1-\alpha) \rceil$-th smallest value of the set of transformed scores $\{T(s_i; \mathcal{S}_C^{(1)}) \mid s_i \in \mathcal{S}_C^{(2)}\}$. Assume that ties among the transformed scores occur with probability zero. Then, denoting by $\mathcal{C}(X') = \{y \in \mathcal{Y} \mid T(s(X', y); \mathcal{S}_C^{(1)}) \leq \hat{t}\}$, $\mathcal{P}(Y' \in \mathcal{C}(X')) \geq 1 - \alpha$, where the probability is defined over the joint draw of the data $\mathcal{D}_C$, $(X', Y')$, and the permutation $\sigma$.*

*Proof.* The overall probability is taken over the joint distribution of the exchangeable data, $\mathcal{D}_C$ and $(X', Y')$, and the independent random permutation, $\sigma$. We use the law of total probability by first conditioning on a specific realization of the permutation, $\sigma = \pi$, and the data in the first split, $\mathcal{D}_C^{(1)} = d^{(1)}$. Given $\sigma = \pi$ and $\mathcal{D}_C^{(1)} = d^{(1)}$, the score set $\mathcal{S}_C^{(1)}$ is fixed. As a result, the function $T(\cdot; \mathcal{S}_C^{(1)})$ becomes a fixed, deterministic transformation.

By the initial exchangeability of all data points, after conditioning on the values of the first split $\mathcal{D}_C^{(1)}$, the remaining $N_{C_2}$ calibration points in $\mathcal{D}_C^{(2)}$ and the test point $(X', Y')$ are still an exchangeable sequence. Applying the fixed transformation $T$ to their scores yields an exchangeable sequence of $N_{C_2} + 1$ scalar values:

$$\{T(s_i; \mathcal{S}_C^{(1)}) \mid (X_i, Y_i) \in \mathcal{D}_C^{(2)}\} \cup \{T(s(X', Y'); \mathcal{S}_C^{(1)})\}$$

Under the no-ties assumption, the rank of the test value $T(s(X', Y'); \mathcal{S}_C^{(1)})$ within this sequence is uniformly distributed on $\{1, \ldots, N_{C_2} + 1\}$. The test point $Y'$ is covered if its transformed score is

less than or equal to the threshold $\hat{t}$. This occurs if and only if the rank of the test score is at most $m = \lceil (N_{C_2} + 1)(1 - \alpha) \rceil$. The probability of this event, conditional on $\sigma = \pi$ and $\mathcal{D}_C^{(1)} = d^{(1)}$, is:

$$\mathcal{P}(Y' \in \mathcal{C}(X') \mid \sigma = \pi, \mathcal{D}_C^{(1)} = d^{(1)}) = \frac{\lceil (N_{C_2} + 1)(1 - \alpha) \rceil}{N_{C_2} + 1} \geq 1 - \alpha.$$

Since this guarantee holds for any realization $(\pi, d^{(1)})$, the unconditional probability also holds by the law of total probability:

$$\mathcal{P}(Y' \in \mathcal{C}(X')) = \mathbb{E}_{\sigma, \mathcal{D}_C^{(1)}} \left[ \mathcal{P}(Y' \in \mathcal{C}(X') \mid \sigma, \mathcal{D}_C^{(1)}) \right] \geq \mathbb{E}_{\sigma, \mathcal{D}_C^{(1)}} [1 - \alpha] = 1 - \alpha,$$

where the expectation is taken over the joint distribution of $\sigma$ and $\mathcal{D}_C^{(1)}$. This completes the proof. □

## C  CSA Algorithm Coverage

We now provide the proof of the coverage guarantees of the region produced by Algorithm 1. As mentioned in the main text, this follows as a direct corollary of Theorem 3.1.

**Corollary C.1.** *Let $\mathcal{D}_C$, $(X', Y')$, $\{s_k\}_{k=1}^K$, and $\alpha$ be as defined in Theorem 3.1. Let $\sigma$, $(\mathcal{S}_C^{(1)}, \mathcal{S}_C^{(2)})$, and $U = \{u_m\}_{m=1}^M$ be as defined by lines 3-4 of the call $\mathrm{CSA}(\{s_k\}, \mathcal{D}_C, 1 - \alpha)$ of Algorithm 1. Denote by $\{\tilde{q}_m\}_{m=1}^M$ the parameters defined by lines 4-9 of Algorithm 1 and by $T$ the scoring function $T(s; \mathcal{S}_C^{(1)}, U) = \max_{m=1,\ldots,M}(u_m^\top s / \tilde{q}_m)$ for any score vector $s \in \mathbb{R}_+^K$. Then, denoting by $\mathcal{C}(X') = \{y \in \mathcal{Y} \mid T(s(X', y); \mathcal{S}_C^{(1)}) \leq \hat{t}\}$, $\mathcal{P}(Y' \in \mathcal{C}(X')) \geq 1 - \alpha$, where the probability is defined over the joint draw of the data $\mathcal{D}_C$, $(X', Y')$, and the permutation $\sigma$.*

*Proof.* To prove the corollary, we must show that this specific function $T$ satisfies the conditions of Theorem 1. The overall probability is taken over the joint draw of the data $(\mathcal{D}_C, (X', Y'))$, the random permutation $\sigma$, and the random directions $U$. We use the law of total probability by conditioning on specific realizations of the random elements $\sigma = \pi$, $\mathcal{D}_C^{(1)} = d^{(1)}$, and $U = u$.

Given these fixed realizations, the score set $\mathcal{S}_C^{(1)}$ and the projection directions $\{u_m\}$ are fixed. The procedure in Algorithm 1 to find the base quantiles $\{\tilde{q}_m\}$ via binary search is a deterministic operation on this fixed data. Therefore, the function $T(s; \mathcal{S}_C^{(1)}, U)$ becomes a fixed, deterministic function of $s$. The conditions of Theorem 1 are met (again assuming no ties in $T$), and its proof implies that the conditional probability of coverage is at least $1 - \alpha$:

$$\mathcal{P}(Y' \in \mathcal{C}(X') \mid \sigma = \pi, \mathcal{D}_C^{(1)} = d^{(1)}, U = u) \geq 1 - \alpha.$$

Since this guarantee holds for any realization $(\pi, d^{(1)}, u)$, the unconditional guarantee follows from the law of total probability:

$$\mathcal{P}(Y' \in \mathcal{C}(X')) = \mathbb{E}_{\sigma, \mathcal{D}_C^{(1)}, U} \left[ \mathcal{P}(Y' \in \mathcal{C}(X') \mid \sigma, \mathcal{D}_C^{(1)}, U) \right] \geq 1 - \alpha.$$

Thus, the guarantee holds for the specific procedure in Algorithm 1. □

## D  CSA Predict-Then-Optimize Visual Walkthrough

We now present a visual accompaniment of the predict-then-optimize algorithm presented in Section 3.2. We once again take $K = 2$ for visual clarity in this walkthrough, where the predictors are as discussed in Section 3.2, namely assumed to be generative predictors $q_k(C \mid X)$ where the number of samples per predictor are fixed to be $\{J_k\}$. For illustration, we assume $J_1 = 5$ and $J_2 = 3$, meaning predictions with the first model are made by drawing 5 samples and 3 for the second. We assume the CSA calibration of Section 3.1 has already been performed, from which a collection of projection directions and quantiles $\{(u_m, \hat{q}_m)\}_{m=1}^M$ are available that implicitly define an acceptance region $\widehat{\mathcal{Q}}$. We further assume the individual predictor score functions are all the GPCP score given in Equation (2), with $d_k$ from Equation (2) specifically here taken to simply be the standard Euclidean 2-norm, giving

$$s_k(x, c) = \min_{j \in 1, \ldots, J_k} ||\hat{c}_{kj} - c||. \tag{4}$$

We now wish to compute $c^* = \max_{\widehat{c} \in \mathcal{C}(x)} f(w, \widehat{c})$. To do so, we must start by defining this region $\mathcal{C}(x)$ for the test point $x$, which we do by drawing the respective number of samples from the two models, producing samples $\{\widehat{c}_{1j}\}_{j=1}^5$ and $\{\widehat{c}_{2j}\}_{j=1}^3$, as shown in Figure 7.

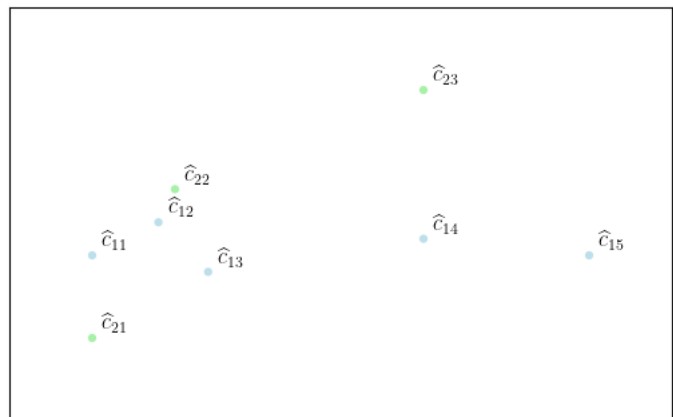

Figure 7: Samples drawn from the two generative models $\{\widehat{c}_{1j}\}_{j=1}^5 \sim q_1(C \mid x)$ (blue) and $\{\widehat{c}_{2j}\}_{j=1}^3 \sim q_2(C \mid x)$ (green). Note that this is a visualization in the $\mathcal{C}$ space, i.e. *not* the space of multivariate scores.

By definition, $\forall c \in \mathcal{C}(x)$,

$$u_m^\top \left( \min_{j_1 = 1, \dots, 5} ||\widehat{c}_{1j_1} - c||, \min_{j_2 = 1, \dots, 3} ||\widehat{c}_{2j_2} - c|| \right) \leq \widehat{q}_m \qquad \forall m = 1, \dots, M. \tag{5}$$

As a result, we must have that, $\forall c \in \mathcal{C}(x)$, $\exists\ j_1 = 1, \dots, 5$ and $j_2 = 1, \dots, 3$ such that $u_m^\top (||\widehat{c}_{1j_1} - c||, ||\widehat{c}_{2j_2} - c||) \leq \widehat{q}_m\ \forall m = 1, \dots, M$. Solving for $c^*$, therefore, amounts to considering each pair $\vec{j} := (j_1, j_2) \in \mathcal{J}$, where $\mathcal{J} := \{1, \dots, 5\} \times \{1, \dots, 3\}$, and solving

$$c_{\vec{j}}^* := \arg\max_c f(w, c)$$
$$\text{s.t.} \quad u_m^\top \left( ||\widehat{c}_{1\vec{j}_1} - c||, ||\widehat{c}_{2\vec{j}_2} - c|| \right) \leq \widehat{q}_m \quad \forall m \in \{1, \dots, M\} \tag{6}$$

Notice that, for any fixed $\vec{j}$, this is a standard convex optimization problem with a convex feasible region. We illustrate how the feasible region would be constructed for a fixed $\vec{j}$ in Figure 8. Note that the construction of this feasible is never explicitly done in practice and is only implicitly used by convex solver routines in practice. We can, therefore, then solve Equation (6) over all possible $\vec{j} \in \mathcal{J}$ and aggregate the maxima to compute $c^*$.

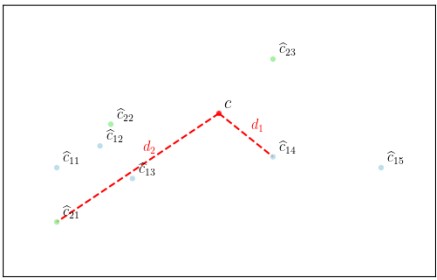
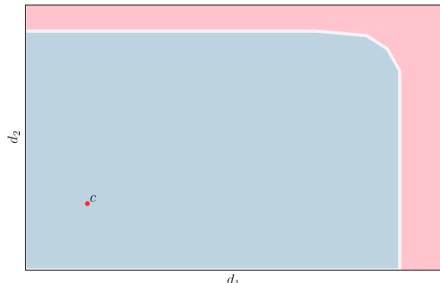

Figure 8: A candidate $c \in \mathcal{C}$ is in prediction region if the projections of its distances $(d_1, d_2)$ from *at least one* pair of points indexed by $\vec{j} := (j_1, j_2)$ is in the acceptance region $\widehat{\mathcal{Q}}$. Here, we illustrate a point $c$ that lies in the feasible region from its proximity to the $\vec{j} := (4, 1)$ pair of points. Note that the left is again a visualization over the $\mathcal{C}$ space, whereas the right is of the *score* space.

# E   Computational Efficiency

## E.1   Vectorized Score Computation

We now discuss the vectorized form of the computations discussed in Section 3.1.2. In particular, putting the scores into a matrix $S_C^{(1)} = [s_1, ..., s_{N_{C_1}}]^\top \in \mathbb{R}^{N_{C_1} \times K}$ and directions into a matrix $U = [u_1, ..., u_M]^\top \in \mathbb{R}^{K \times M}$, all the projections $u_m^\top s_i$ can be computed as $S_C^{(1)} U \in \mathbb{R}^{N_{C_1} \times M}$, where $[S_C^{(1)} U]_{i,m}$ is precisely $u_m^\top s_i$. $\tilde{q} \in \mathbb{R}^M := \{\tilde{q}_m := \text{quantile}([S_C^{(1)} U]_{:,m}; 1 - \alpha)\}$ is then the quantile per row.

For any test point $s' \in \mathbb{R}^K$, we can then very efficiently check if it falls into the region by checking if it satisfies $Us' \leq \tilde{q}$ component-wise. Each iteration of the loop to find $\beta^*$, therefore, is very fast, and we find the search typically converges in 5-10 iterations.

The final step is computing $\widehat{t}$. Computing this follows similarly to above, where we take the scores $S_C^{(2)}$, compute projections $S_C^{(2)} U \in \mathbb{R}^{N_{C_2} \times M}$, find $\widetilde{T} := S_C^{(2)} U / \tilde{q} \in \mathbb{R}^{N_{C_2} \times M}$, where division is interpreted as being defined component-wise along the rows, computing the maxima similarly along the rows $[T^*]_i := \max \widetilde{T}_{i,:}$, and finally computing $\widehat{t}$ as the $1 - \alpha$ quantile of $T^*$.

## E.2   Empirical Efficiency Validation

To demonstrate the computational efficiency of this vectorized approach, we reran the experiment on the "Parkinsons" UCI task with both varying numbers of predictors ($K$) and projection directions ($M$); for each combination, we measured the total time taken to compute the quantile (i.e. to run Algorithm 1) and to perform the projection to assess coverage for the test points. The additional predictors were taken to be random forests with different numbers of trees. $K$ is given in the left column and $M$ in each column heading, with the entry for each $(K, M)$ pair being reported in seconds. As expected, by the vectorized nature of the computations, as discussed in the main paper, the performance scales gracefully over $M$ at roughly $\mathcal{O}(M)$ and remains roughly constant in $K$.

Table 4:  Performance values for varying $K$ (number of predictors) and $M$ (number of projection directions). All values are reported in seconds.

| $K$ \ $M$ | 10 | 100 | 1000 | 10000 |
|---|---|---|---|---|
| 6 | 0.111668 | 0.373029 | 2.32803 | 37.2561 |
| 8 | 0.0961056 | 0.327051 | 2.16211 | 36.7216 |
| 10 | 0.117146 | 0.373464 | 2.73875 | 37.2603 |
| 12 | 0.123772 | 0.384527 | 2.35386 | 37.0735 |

# F   Prediction Region Intuition

We now consider a simplified setting of the general procedure to gain insight into the efficiency resulting prediction regions. In particular, we consider a scalar regression setting with $K$ predictors $f_1, ..., f_K : \mathcal{X} \to \mathbb{R}$. We further suppose, with a slight abuse of notation, $f(x) - y \sim \mathcal{N}(0, \Sigma)$ for $f(x) := [f_1(x), ..., f_K(x)]$ and that the scores are taken to be $s_k(x, y) := (f_k(x) - y)^2$. Then, as $\widehat{\mathcal{Q}}$ is precisely the region with minimal volume that captures $1 - \alpha$ density, it is precisely given by $\chi^2_{K,1-\alpha}$, the $1 - \alpha$ quantile of the $\chi^2$ distribution with $K$ degrees of freedom. That is, the prediction regions are $\mathcal{C}^{\text{CSA}}(x) := \{y : (f(x) - y)^\top \Sigma^{-1} (f(x) - y) \leq \chi^2_{K,1-\alpha}\}$. Notably, if $\Sigma = \text{diag}(\{\sigma_i^2\})$,

$$\mathcal{C}^{\text{CSA-diag}}(x) := \left\{ y : \sum_{i=1}^{K} \left( \frac{f_k(x) - y}{\sigma_k} \right)^2 \leq \chi^2_{K,1-\alpha} \right\}. \tag{7}$$

Two insights arise from this expression. The first is that, unlike in the prediction regions for the case of the univariate score function $s_k(x, y) := (f_k(x) - y)^2$, the size $|\mathcal{C}^{\text{CSA-diag}}(x)|$ is *dependent* on $x$. In the univariate case, the size is $2\widehat{q}_k$ across all $x$. Here, however, the feasible set of $y$ becomes smaller the more distinct the values $f_k(x)$ are. The second insight, therefore, is that, under such independence of residuals, prediction region sizes are minimized in having well-separated predictions, which suggests that efficiency is optimized by having an ensemble of predictors that learn distinct maps from $\mathcal{X} \to \mathcal{Y}$, such as those that focus on distinct views of the covariate space.

# G   Full ImageNet Results

Here we present the complete collection of results for the classification task across additional $\alpha$ than those that could fit in the main paper.

Table 5:  Average coverages across different coverage levels are shown in the top rows and average prediction set sizes in the bottom rows. Both were assessed over a batch of i.i.d. test samples (15% of the validation set from ImageNet). Standard deviations and means were computed across 10 randomized draws of the calibration and test sets.

| Dataset/$\alpha$ | Metrics | ResNet | VGG | DenseNet | VFCP | $\mathcal{C}^M$ | $\mathcal{C}^R$ | $\mathcal{C}^U$ | Ensemble | CSA (Single-Stage) | CSA |
|---|---|---|---|---|---|---|---|---|---|---|---|
| ImageNet | Coverage | 0.97 (0.011) | 0.966 (0.011) | 0.939 (0.002) | 0.954 (0.006) | 0.956 (0.011) | 0.948 (0.01) | 0.96 (0.013) | 0.95 (0.006) | 0.955 (0.013) | 0.957 (0.01) |
| ($\alpha = 0.07$) | Length | 174.828 (2.037) | 181.58 (2.459) | 160.623 (2.828) | 61.247 (1.638) | 122.899 (2.154) | 111.602 (2.155) | 173.264 (2.674) | 86.955 (1.886) | 44.787 (1.198) | 45.352 (1.56) |
| ($\alpha = 0.05$) | Coverage | 0.95 (0.003) | 0.949 (0.004) | 0.952 (0.002) | 0.95 (0.003) | 0.975 (0.002) | 0.954 (0.004) | 0.95 (0.003) | 0.949 (0.002) | 0.95 (0.002) | 0.95 (0.003) |
| | Length | 220.022 (2.072) | 229.523 (3.076) | 208.658 (2.016) | 78.108 (2.004) | 166.933 (2.157) | 143.323 (2.932) | 220.491 (2.773) | 112.161 (2.115) | 58.424 (1.674) | 59.574 (3.382) |
| ($\alpha = 0.02$) | Coverage | 0.98 (0.002) | 0.981 (0.002) | 0.98 (0.002) | 0.98 (0.002) | 0.992 (0.002) | 0.982 (0.002) | 0.98 (0.002) | 0.98 (0.003) | 0.98 (0.001) | 0.979 (0.002) |
| | Length | 357.487 (5.046) | 363.376 (4.916) | 342.479 (5.603) | 137.356 (3.595) | 311.521 (4.053) | 327.754 (6.251) | 355.701 (4.423) | 202.573 (2.948) | 117.272 (5.015) | 121.477 (19.719) |
| ($\alpha = 0.01$) | Coverage | 0.99 (0.001) | 0.991 (0.001) | 0.989 (0.002) | 0.99 (0.001) | 0.997 (0.001) | 0.991 (0.002) | 0.99 (0.002) | 0.99 (0.002) | 0.99 (0.001) | 0.99 (0.002) |
| | Length | 491.952 (6.353) | 726.028 (12.157) | 459.399 (6.739) | 194.691 (4.579) | 580.592 (7.715) | 532.155 (24.829) | 559.188 (7.07) | 299.453 (6.526) | 180.534 (8.468) | 201.32 (46.509) |

# H   Full OpenML Results

Table 6:  Average coverages across tasks for $\alpha = 0.05$ are shown in the top row and average prediction set lengths in the bottom row, where both were assessed over a batch of i.i.d. test samples (20% of the dataset size). Standard deviations and means were computed across 5 randomizations of draws of the training, calibration, and test sets. In cases where the method failed to achieve sufficient coverage (defined as $< 0.93$), we do not include it in comparison for set length. Similarly, the single-stage approach fails to achieve coverage due to lack of exchangeability with test points.

| Dataset | Metrics | Linear Model | LASSO | Random Forest | XGBoost | $\mathcal{C}^M$ | $\mathcal{C}^R$ | $\mathcal{C}^U$ | Ensemble | CSA (Single-Stage) | CSA |
|---|---|---|---|---|---|---|---|---|---|---|---|
| 361234 | Coverage | 0.97 (0.011) | 0.966 (0.011) | 0.939 (0.002) | 0.954 (0.006) | 0.956 (0.011) | 0.948 (0.01) | 0.96 (0.013) | 0.95 (0.006) | 0.955 (0.013) | 0.957 (0.01) |
| | Length | 9.673 (0.160) | 9.645 (0.154) | 10.080 (0.160) | 9.157 (0.052) | 9.196 (0.123) | 8.703 (0.086) | 9.524 (0.056) | 17.759 (0.275) | 7.646 (0.073) | 7.688 (0.181) |
| 361235 | Coverage | 0.947 (0.0) | 0.945 (0.005) | 0.968 (0.016) | 0.95 (0.005) | 0.955 (0.016) | 0.953 (0.011) | 0.897 (0.005) | 0.932 (0.021) | 0.745 (0.011) | 0.984 (0.005) |
| | Length | 20.961 (0.651) | 24.241 (0.246) | 10.096 (0.587) | 11.387 (0.452) | 11.782 (0.057) | — | 16.088 (0.118) | 15.823 (1.272) | 6.162 (0.458) | 11.695 (0.266) |
| 361236 | Coverage | 0.975 (0.008) | 0.975 (0.008) | 0.961 (0.0) | 0.948 (0.012) | 0.948 (0.012) | 0.938 (0.012) | 0.965 (0.008) | 0.934 (0.004) | 0.94 (0.004) | 0.963 (0.004) |
| | Length | 44407.071 (1173.758) | 44509.269 (1229.817) | 50820.568 (385.951) | 41045.069 (1221.808) | 43185.942 (1002.516) | 40905.295 (1089.411) | 44437.938 (851.862) | 60509.250 (2410.320) | 30953.589 (2482.065) | 33439.322 (1275.213) |
| 361237 | Coverage | 0.969 (0.023) | 0.969 (0.023) | 0.981 (0.0) | 0.923 (0.0) | 0.954 (0.015) | 0.9 (0.008) | 0.969 (0.023) | 0.885 (0.038) | — | 0.977 (0.008) |
| | Length | 44.019 (0.990) | 44.069 (1.115) | 27.035 (1.014) | — | 26.524 (1.244) | — | 31.967 (1.118) | — | 14.473 (0.503) | 23.145 (0.199) |
| 361241 | Coverage | 0.954 (0.001) | 0.956 (0.001) | 0.944 (0.005) | 0.957 (0.002) | 0.954 (0.002) | 0.923 (0.0) | 0.952 (0.0) | 0.949 (0.001) | 0.917 (0.006) | 0.951 (0.001) |
| | Length | 19.133 (0.062) | 20.245 (0.095) | 18.102 (0.055) | 18.482 (0.062) | 17.958 (0.062) | — | 18.932 (0.034) | 29.548 (0.191) | 15.199 (0.427) | 17.328 (0.097) |
| 361242 | Coverage | 0.944 (0.004) | 0.955 (0.0) | 0.947 (0.004) | 0.942 (0.0) | 0.948 (0.0) | 0.914 (0.003) | 0.944 (0.001) | 0.949 (0.003) | 0.9 (0.006) | 0.944 (0.002) |
| | Length | 70.248 (0.304) | 84.510 (0.282) | 50.442 (0.421) | 54.844 (0.036) | 54.217 (0.115) | — | 65.635 (0.094) | 61.613 (0.372) | 44.602 (0.170) | 57.935 (0.070) |
| 361243 | Coverage | 0.922 (0.03) | 0.952 (0.015) | 0.956 (0.022) | 0.952 (0.015) | 0.937 (0.022) | 0.893 (0.044) | 0.937 (0.022) | 0.919 (0.022) | 0.748 (0.126) | 0.956 (0.022) |
| | Length | — | 71.388 (0.152) | 75.924 (2.291) | 72.877 (0.729) | 68.493 (0.993) | — | 72.048 (0.024) | — | 43.742 (10.285) | 68.220 (1.605) |
| 361244 | Coverage | 0.97 (0.022) | 0.97 (0.022) | 0.97 (0.022) | 0.97 (0.022) | 0.97 (0.022) | 0.97 (0.022) | 0.97 (0.022) | 0.974 (0.015) | 0.963 (0.037) | 0.956 (0.015) |
| | Length | 3.274 (0.004) | 3.274 (0.004) | 3.336 (0.023) | 3.284 (0.010) | 3.272 (0.000) | 3.269 (0.003) | 3.289 (0.002) | 4.854 (0.293) | 0.287 (0.008) | 0.287 (0.008) |
| 361247 | Coverage | 0.96 (0.001) | 0.953 (0.003) | 0.94 (0.001) | 0.951 (0.003) | 0.963 (0.001) | 0.903 (0.007) | 0.951 (0.0) | 0.954 (0.001) | 0.843 (0.003) | 0.943 (0.006) |
| | Length | 0.025 (0.000) | 0.038 (0.000) | 0.006 (0.000) | 0.016 (0.000) | 0.015 (0.000) | — | 0.022 (0.000) | 0.013 (0.000) | 0.005 (0.000) | 0.008 (0.000) |
| 361249 | Coverage | 0.96 (0.002) | 0.956 (0.002) | 0.962 (0.003) | 0.972 (0.007) | 0.953 (0.005) | 0.938 (0.002) | 0.965 (0.005) | 0.936 (0.01) | 0.931 (0.0) | 0.953 (0.002) |
| | Length | 3.008 (0.006) | 3.068 (0.009) | 2.800 (0.000) | 2.780 (0.025) | 2.775 (0.006) | 2.558 (0.019) | 2.894 (0.000) | 4.706 (0.099) | 2.216 (0.020) | 2.614 (0.043) |

Table 7:  Average coverages across tasks for $\alpha = 0.025$ are shown in the top row and average prediction set lengths in the bottom row, where both were assessed over a batch of i.i.d. test samples (20% of the dataset size). Standard deviations and means were computed across 5 randomizations of draws of the training, calibration, and test sets. In cases where the method failed to achieve sufficient coverage (defined as $< 0.96$), we do not include it in comparison for set length. Similarly, the single-stage approach fails to achieve coverage due to lack of exchangeability with test points.

| Dataset | Metrics | Linear Model | LASSO | Random Forest | XGBoost | $\mathcal{C}^M$ | $\mathcal{C}^R$ | $\mathcal{C}^U$ | Ensemble | CSA (Single-Stage) | CSA |
|---|---|---|---|---|---|---|---|---|---|---|---|
| 361234 | Coverage | 0.987 (0.008) | 0.987 (0.008) | 0.974 (0.004) | 0.977 (0.008) | 0.982 (0.008) | 0.971 (0.01) | 0.981 (0.01) | 0.97 (0.008) | 0.976 (0.01) | 0.973 (0.006) |
| | Length | 11.939 (0.137) | 11.871 (0.084) | 12.484 (0.168) | 11.972 (0.009) | 11.587 (0.110) | 11.157 (0.086) | 11.965 (0.050) | 25.598 (0.974) | 9.306 (0.259) | 8.855 (0.059) |
| 361235 | Coverage | 0.987 (0.0) | 0.982 (0.011) | 0.979 (0.011) | 0.984 (0.005) | 0.989 (0.005) | 0.966 (0.014) | 0.976 (0.005) | 0.958 (0.021) | 0.889 (0.011) | 0.989 (0.005) |
| | Length | 24.595 (0.825) | 28.841 (1.129) | 11.811 (0.992) | 14.237 (0.786) | 14.472 (0.172) | 12.278 (0.026) | 19.231 (0.356) | — | 7.719 (0.467) | 12.563 (0.766) |
| 361236 | Coverage | 0.992 (0.004) | 0.992 (0.004) | 0.981 (0.0) | 0.965 (0.008) | 0.975 (0.008) | 0.965 (0.008) | 0.977 (0.012) | 0.955 (0.008) | 0.955 (0.012) | 0.973 (0.004) |
| | Length | 48591.496 (873.946) | 48578.821 (867.718) | 56132.760 (368.832) | 46623.663 (1565.744) | 47630.890 (810.373) | 45714.911 (1062.420) | 49188.464 (753.205) | — | 32881.580 (3110.734) | 35777.096 (1949.538) |
| 361237 | Coverage | 0.981 (0.0) | 0.981 (0.0) | 0.981 (0.0) | 0.977 (0.008) | 0.962 (0.0) | 0.962 (0.0) | 0.977 (0.008) | 0.965 (0.008) | 0.927 (0.031) | 0.981 (0.0) |
| | Length | 47.738 (0.542) | 47.440 (0.959) | 30.785 (0.037) | 26.208 (0.897) | 30.554 (0.561) | 27.182 (0.803) | 35.982 (0.619) | 67.660 (6.380) | 18.214 (0.436) | 26.897 (0.515) |
| 361241 | Coverage | 0.979 (0.001) | 0.978 (0.001) | 0.976 (0.001) | 0.978 (0.001) | 0.978 (0.0) | 0.964 (0.002) | 0.977 (0.0) | 0.972 (0.002) | 0.958 (0.003) | 0.979 (0.0) |
| | Length | 21.772 (0.085) | 23.089 (0.106) | 21.543 (0.009) | 21.454 (0.109) | 20.862 (0.088) | 19.291 (0.060) | 21.905 (0.041) | 40.082 (0.045) | 17.765 (0.329) | 19.897 (0.062) |
| 361242 | Coverage | 0.977 (0.003) | 0.978 (0.001) | 0.975 (0.002) | 0.968 (0.003) | 0.973 (0.004) | 0.955 (0.002) | 0.971 (0.002) | 0.975 (0.0) | 0.936 (0.001) | 0.977 (0.001) |
| | Length | 83.892 (0.143) | 99.811 (0.866) | 65.672 (0.212) | 68.119 (0.187) | 68.155 (0.357) | — | 80.032 (0.354) | 85.678 (0.371) | 53.549 (0.585) | 69.388 (0.128) |
| 361243 | Coverage | 0.985 (0.007) | 0.985 (0.007) | 0.985 (0.007) | 0.956 (0.022) | 0.985 (0.007) | 0.97 (0.015) | 0.985 (0.007) | 0.978 (0.008) | 0.748 (0.126) | 0.985 (0.007) |
| | Length | 92.698 (1.567) | 87.949 (0.147) | 88.993 (2.345) | — | 84.569 (0.557) | 79.879 (0.739) | 87.950 (0.308) | 137.673 (15.214) | 46.504 (12.957) | 79.976 (2.585) |
| 361244 | Coverage | 0.974 (0.015) | 0.974 (0.015) | 0.974 (0.015) | 0.974 (0.015) | 0.974 (0.015) | 0.974 (0.015) | 0.974 (0.015) | 0.989 (0.022) | 0.963 (0.037) | 0.974 (0.015) |
| | Length | 5.274 (0.004) | 5.274 (0.004) | 5.336 (0.023) | 5.284 (0.010) | 5.272 (0.000) | 5.269 (0.003) | 5.289 (0.002) | 11.283 (1.039) | 0.287 (0.008) | 0.287 (0.008) |
| 361247 | Coverage | 0.98 (0.0) | 0.976 (0.003) | 0.969 (0.004) | 0.974 (0.001) | 0.977 (0.003) | 0.945 (0.004) | 0.971 (0.003) | 0.982 (0.000) | 0.906 (0.003) | 0.974 (0.004) |
| | Length | 0.029 (0.000) | 0.042 (0.000) | 0.009 (0.000) | 0.019 (0.000) | 0.018 (0.000) | — | 0.025 (0.000) | 0.016 (0.000) | 0.008 (0.000) | 0.012 (0.000) |
| 361249 | Coverage | 0.981 (0.003) | 0.981 (0.003) | 0.984 (0.002) | 0.991 (0.002) | 0.981 (0.003) | 0.976 (0.002) | 0.981 (0.003) | 0.971 (0.007) | 0.962 (0.011) | 0.977 (0.003) |
| | Length | 3.645 (0.023) | 3.674 (0.026) | 3.600 (0.000) | 3.322 (0.019) | 3.402 (0.005) | 3.201 (0.019) | 3.543 (0.000) | 6.533 (0.252) | 2.719 (0.164) | 2.972 (0.064) |

# I  CSA Region Size Over $M$

Across all the choices of $M$ presented in the below table, the coverages were identical, namely 0.981 for the $\alpha = 0.025$ case and 0.962 for $\alpha = 0.05$ (for task 361237).

Table 8: Comparison of interval lengths for different $M$ values.

| $M$ | Length ($\alpha = 0.025$) | Length ($\alpha = 0.05$) |
|---|---|---|
| 50 | 28.037 | 23.017 |
| 100 | 27.111 | 22.071 |
| 500 | 25.880 | 22.621 |
| 1000 | 26.110 | 22.172 |
| 5000 | 24.943 | 22.037 |

# J  CSA Prediction Region Visualizations

We now visualize some of the prediction regions corresponding to some of the trials run in Appendix H. While we find these intervals to be connected across these tasks, we expect visualizations over multivariate output spaces, i.e. for 2D regression problems, would reveal sets to be non-connected.

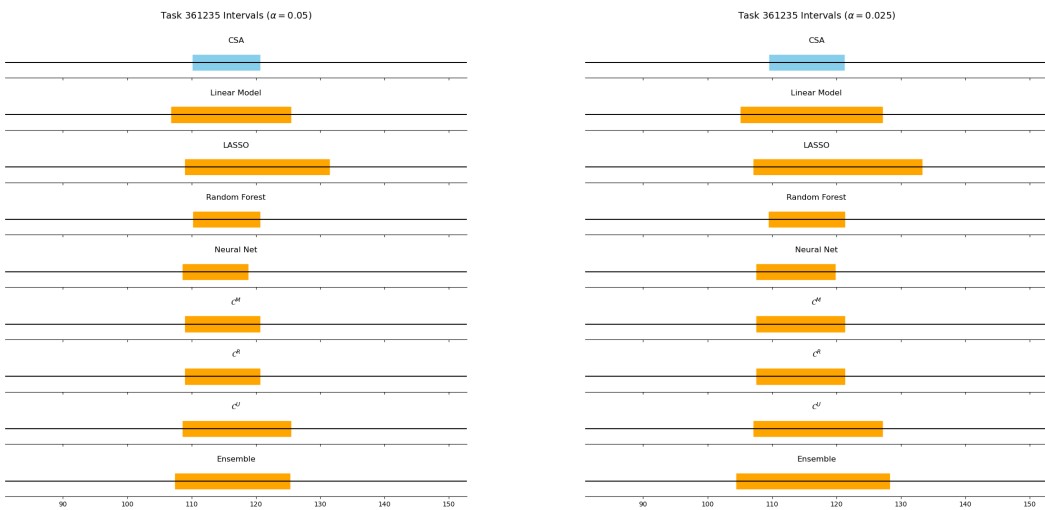

Figure 9: Prediction regions across methods for task 361235 for $\alpha = 0.05$ (left) and $0.025$ (right).

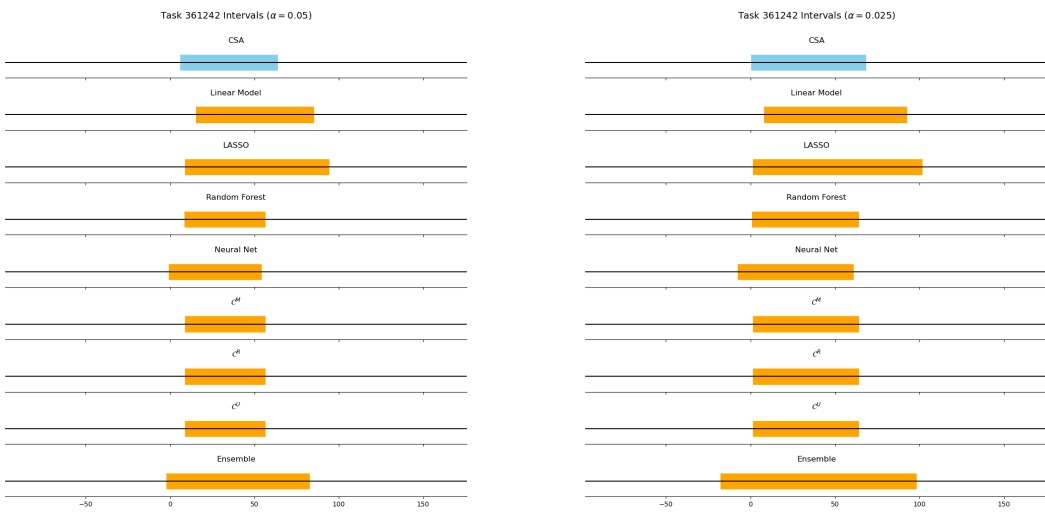

Figure 10: Prediction regions across methods for task 361242 for $\alpha = 0.05$ (left) and $0.025$ (right).

# K    UCI Results

We consider those regression tasks from the UCI repository [44] that have at least 1,000 samples. The complete collection of results is presented in Table 9. As discussed in the main text, across nearly all the UCI benchmark tasks, we find that the conformalized random forest does optimally and that ensembling methods provide no further benefit over simply taking this single predictor. In this degenerate case, we would expect an optimal aggregator to simply then return this optimal single predictor. We then find CSA to consistently significantly outperform other aggregation strategies and to return a prediction region of size comparable to that of the nominal random forest.

Table 9: Average coverages across tasks for $\alpha = 0.05$ are shown in the top row and average prediction set lengths in the bottom row, where both were assessed over a batch of i.i.d. test samples (10% of the dataset size). We are highlighting the robustness compared to other aggregation strategies here and so bold the best performing amongst the aggregation methods. Standard deviations and means were computed across 5 randomizations of draws of the training, calibration, and test sets. Note that, while the single-stage prediction regions are the smallest, they fail to achieve the desired coverage level and are, therefore, precluded from comparison.

| Dataset | Metrics | Linear Model | LASSO | Random Forest | XGBoost | $\mathcal{C}^M$ | $\mathcal{C}^R$ | $\mathcal{C}^U$ | Ensemble | CSA (Single-Stage) | CSA |
|---|---|---|---|---|---|---|---|---|---|---|---|
| airfoil | Coverage | 0.966 (0.011) | 0.926 (0.011) | 0.934 (0.026) | 0.966 (0.011) | 0.945 (0.021) | 0.916 (0.016) | 0.934 (0.026) | 0.958 (0.032) | 0.805 (0.058) | 0.953 (0.011) |
| | Length | 19.062 (0.615) | — | 11.368 (0.188) | 11.770 (0.261) | **12.371 (0.131)** | 16.227 (0.025) | 16.097 (1.052) | 16.097 (1.052) | 8.609 (0.719) | 14.075 (0.734) |
| bike | Coverage | 0.944 (0.007) | 0.947 (0.004) | 0.954 (0.002) | 0.958 (0.003) | 0.938 (0.006) | 0.908 (0.0) | 0.946 (0.003) | 0.955 (0.0) | 0.963 (0.007) | 0.947 (0.003) |
| | Length | 3.178 (0.011) | 3.343 (0.021) | 0.065 (0.000) | 0.111 (0.000) | **0.111 (0.001)** | 0.111 (0.001) | 1.722 (0.007) | 0.682 (0.004) | 0.156 (0.019) | 0.134 (0.007) |
| concrete | Coverage | 0.977 (0.008) | 0.977 (0.008) | 0.981 (0.0) | 0.915 (0.023) | 0.962 (0.0) | 0.915 (0.023) | 0.981 (0.0) | 0.915 (0.023) | 0.854 (0.054) | 0.977 (0.008) |
| | Length | 44.295 (0.424) | 44.470 (0.326) | 26.053 (2.114) | — | **26.488 (1.141)** | — | 32.455 (1.165) | — | 19.712 (6.592) | **25.302 (3.244)** |
| kin40k | Coverage | 0.949 (0.002) | 0.949 (0.001) | 0.946 (0.002) | 0.948 (0.003) | 0.945 (0.002) | 0.919 (0.002) | 0.949 (0.0) | 0.945 (0.002) | 0.907 (0.004) | 0.941 (0.006) |
| | Length | 3.781 (0.017) | 3.781 (0.016) | 2.333 (0.026) | 3.343 (0.026) | 3.269 (0.018) | — | 3.291 (0.009) | 4.985 (0.003) | 2.138 (0.001) | **2.456 (0.042)** |
| parkinsons | Coverage | 0.937 (0.003) | 0.946 (0.0) | 0.958 (0.009) | 0.953 (0.007) | 0.936 (0.001) | 0.904 (0.008) | 0.936 (0.005) | 0.955 (0.004) | 0.873 (0.043) | 0.951 (0.003) |
| | Length | 35.957 (0.323) | 36.430 (0.328) | 3.254 (0.423) | 11.268 (0.114) | 10.992 (0.074) | — | 21.470 (0.003) | 14.161 (0.083) | 3.457 (0.727) | **4.584 (0.637)** |
| pol | Coverage | 0.944 (0.001) | 0.942 (0.002) | 0.951 (0.004) | 0.955 (0.001) | 0.938 (0.001) | 0.909 (0.003) | 0.946 (0.001) | 0.953 (0.005) | 0.884 (0.009) | 0.952 (0.003) |
| | Length | 97.944 (0.150) | 97.771 (0.386) | 28.000 (0.000) | 48.572 (1.279) | 45.056 (0.821) | — | 69.078 (0.362) | 57.432 (0.663) | 24.321 (1.370) | **33.230 (1.569)** |
| protein | Coverage | 0.958 (0.001) | 0.957 (0.002) | 0.953 (0.003) | 0.959 (0.003) | 0.957 (0.003) | 0.928 (0.003) | 0.953 (0.003) | 0.955 (0.0) | 0.91 (0.002) | 0.963 (0.004) |
| | Length | 2.316 (0.002) | 2.412 (0.011) | 2.151 (0.014) | 2.210 (0.006) | 2.134 (0.002) | — | 2.269 (0.001) | 3.707 (0.039) | 1.717 (0.036) | **1.994 (0.000)** |
| pumadyn32nm | Coverage | 0.961 (0.011) | 0.961 (0.01) | 0.947 (0.001) | 0.962 (0.003) | 0.96 (0.007) | 0.935 (0.003) | 0.95 (0.013) | 0.957 (0.013) | 0.906 (0.026) | 0.963 (0.001) |
| | Length | 3.997 (0.024) | 3.979 (0.031) | 1.525 (0.027) | 3.518 (0.058) | 3.507 (0.051) | 2.350 (0.043) | 3.242 (0.033) | 5.351 (0.139) | 1.564 (0.048) | **1.858 (0.078)** |
| tamielectric | Coverage | 0.953 (0.002) | 0.953 (0.002) | 0.947 (0.003) | 0.953 (0.003) | 0.952 (0.003) | 0.926 (0.007) | 0.949 (0.0) | 0.948 (0.003) | 0.909 (0.003) | 0.949 (0.002) |
| | Length | 0.950 (0.001) | 0.951 (0.001) | 1.271 (0.006) | 0.953 (0.001) | 0.948 (0.001) | — | 1.029 (0.003) | 4.539 (0.038) | 0.774 (0.005) | **0.799 (0.004)** |
| wine | Coverage | 0.96 (0.005) | 0.943 (0.01) | 0.931 (0.015) | 0.923 (0.02) | 0.928 (0.005) | 0.884 (0.015) | 0.948 (0.005) | 0.946 (0.015) | 0.805 (0.054) | 0.933 (0.015) |
| | Length | 2.352 (0.002) | 3.521 (0.025) | 2.619 (0.050) | — | — | — | 2.621 (0.039) | 3.390 (0.042) | 1.683 (0.224) | **2.291 (0.026)** |

# L    Robust Traffic Routing Setup

We replicate the experimental setup of [36], namely where a graph of Manhattan with corresponding nominal transit times was extracted using OSMnx [49]. Formally, the Manhattan graph is given as a tuple $(\mathcal{V}, \mathcal{E})$, where the edge weights represent the transit times along the respective city roads.

Such weights were assigned in a two-step process, namely by first making weather predictions and then using such weather predictions to then upweight the nominal transit times. In particular, precipitation forecasts were made from time-series observations of previous precipitations readings, specifically given over a map spatially resolved to $H \times W$ resolution. Precipitation forecasters, such as those considered in the experiments herein as given in [45] and [46], specifically map such previous observations to potential future trajectories. Formally, they define probabilistic models over some future time horizon $T_f$, from which probabilistic draws $\widetilde{Y} \in \mathbb{R}^{T_f \times H \times W} \sim \mathcal{P}(\widetilde{Y} \mid x)$ can be made, where $x \in \mathbb{R}^{T \times H \times W}$. Notably, we instead consider the probabilistic forecasts at some future *fixed* time point $T'$, meaning the outcome of interest $Y \in \mathbb{R}^{H \times W} = \widetilde{Y}_{T'}$

From a precipitation map, namely a spatially resolved reading $Y \in \mathbb{R}^{H \times W}$, we assign the final edge weights by first associating nodes to the closest pixel coordinate of the precipitation map. That is, denoting the pixel nearest to a vertex $v$ as $(p_x^v, p_y^v)$, the node is assigned the value at such a spatial location $Y_{p_y^v, p_x^v}$. To, therefore, assign the edge weight, we average the weights of the edge endpoints and then weigh the nominal transit time. In particular, denoting the nominal transit time along such an edge $e$ between nodes $(s, t)$ as $\widetilde{c}_e$, the transit time with traffic was computed as

$$c_e := \widetilde{c}_e \cdot \exp\left\{\frac{Y_{p_x^{e_s}, p_y^v} + Y_{p_x^{e_t}, p_y^{e_t}}}{2}\right\}. \tag{8}$$

## M    Compute Details

All OpenML were all run on a standard-grade CPU. The deep learning-based experiments, namely the ImageNet classification and traffic forecasting predict-then-optimize task, were performed on an Nvidia RTX 2080 Ti GPU. Such experiments, however, were conducted with publicly available, pre-trained models provided by the works respectively referenced in the sections describing the experimental setups.

## N    Conformal Aggregation Methods

We now describe the methods from [25] that were compared against experimentally, specifically the standard majority-vote $\mathcal{C}^M$, partially randomized thresholding $\mathcal{C}^R$, and fully randomized thresholding $\mathcal{C}^U$ approaches. As discussed in Section 2.4, these methods all follow the structural form of

$$\mathcal{C}(x) := \left\{ y \mid \sum_{k=1}^{K} w_k \mathbb{1}[y \in \mathcal{C}_k(x)] \geq \widehat{a} \right\} \tag{9}$$

and largely differ in their choice of weights and thresholds. The standard majority-vote $\mathcal{C}^M$ is the most natural choice, defined by

$$\mathcal{C}^M(x) := \left\{ y \mid \frac{1}{K} \sum_{k=1}^{K} \mathbb{1}[y \in \mathcal{C}_k(x)] > \frac{1}{2} \right\}. \tag{10}$$

The randomized methods differ in that independent randomization is leveraged over the threshold, namely with:

$$\mathcal{C}^R(x) := \left\{ y \mid \frac{1}{K} \sum_{k=1}^{K} \mathbb{1}[y \in \mathcal{C}_k(x)] > \frac{1}{2} + \frac{U}{2} \right\} \tag{11}$$

$$\mathcal{C}^U(x) := \left\{ y \mid \frac{1}{K} \sum_{k=1}^{K} \mathbb{1}[y \in \mathcal{C}_k(x)] > U \right\}, \tag{12}$$

for $U \sim \mathrm{Unif}([0, 1])$. Notably, all these methods retain the guarantees typical of conformal prediction.

