# OpenReview forum: "Conformal Prediction for Ensembles: Improving Efficiency via Score-Based Aggregation"
_NeurIPS.cc/2025/Conference — NeurIPS 2025 poster_

### Official Review · Reviewer_RKaG · 2025-06-23

**Clarity:** 3
**Significance:** 3
**Originality:** 3
**Rating:** 5
**Confidence:** 2

**Summary:**

Authors propose conformal score aggregation (CSA) a new method for adapting conformal prediction to ensembles. Instead of averaging the scores of each model, they construct a data-driven ordering of the score vectors. This pre-ordering is achieved using a set of nested sets, each defined by a threshold parameter. Once the conformal scores are ordered, the coverage guarantee can be computed. Then, they go over a binary search over a separate parameter to obtain the desired coverage based on that ordering. They prove the theoretical consistency of CSA and provide experiments demonstrating the improvements it offers.

**Questions:**

Q1. The conclusions section appears as Discussion, it should be self-contained and have more detail.

Q2. The paper can be improved with a comment on the limitations of CSA.

Q3. We are not sufficiently familiar with this specific field to provide a confident assessment, however the papper seems technically sound and consistent. This is spetially the case for sections 3.2 and 4.2.

**Ethical Concerns:**

["NO or VERY MINOR ethics concerns only"]

**Final Justification:**

The concerns raised in the review were properly addressed by the authors. Since these were minor comments, I will keep my rating as it was.

**Limitations:**

No, the authors should comment on limitations of the proposed method.

**Paper Formatting Concerns:**

No major formatting issues.

**Quality:**

3

**Strengths And Weaknesses:**

Strengths:

S1. The CSA framework is novel and takes a different approach on the adaptation of conformal prediction to ensembles. (Originality).

S2. CSA experimentally proves to yield significantly less conservative regions than previous methods. (Significance).

S3. Computational efficiency.

S4. The writing is clear and the exposition and methods are sound. (Quality, Clarity).

Weaknesses:

W1. There is no comment on the limitations of the proposed method.

W2. The article lacks a conclusion section.

Minor comments:
E1. Line 230 grammatical error.

---

> ### Author Rebuttal · Authors · 2025-07-29
>
> We thank the reviewer for their time and questions. We address each of the points successively below.
>
> **The conclusions section appears as Discussion, it should be self-contained and have more detail.**
>
> We will expand the discussion in the final submission to include more details about extensions that can be made on the current CSA framework.
>
> **The paper can be improved with a comment on the limitations of CSA.**
>
> We will extend the conclusions to note some limitations. Some current ones were highlighted in the discussion section, where it would be of interest to see further use cases of CSA being integrated into regression settings beyond the predict-then-optimize setting we considered in the paper. Another limitation might be in the loss of data efficiency in needing to further split the calibration data into $D_ {C_1}$ and $D_ {C_2}$ for determining the quantile envelope shape without sacrificing coverage guarantees. Similar to how there is full conformal prediction as a way to augment split conformal use cases when it is infeasible to hold out a separate calibration set, it could likely be possible to generate a similar “full CSA” procedure, where the set is not split and where we sacrifice computational efficiency for data efficiency.

---

> > ### Comment · Reviewer_RKaG · 2025-08-06
> > **Answer to Rebuttal**
> >
> > I appreciate the answer of the authors. My concerns are addressed since the discussion will be expanded and some limitations have been explained.
> > I will keep my score as it was.

---

### Official Review · Reviewer_xX66 · 2025-06-29

**Clarity:** 3
**Significance:** 3
**Originality:** 4
**Rating:** 5
**Confidence:** 4

**Summary:**

The authors propose a method to merge prediction sets from different predictors, each being a valid prediction set with at least $1-\alpha$ coverage rate. They take advantage of the quantile envelope idea but make substantial changes. Most importantly, they develop a nontrivial method to correct the coverage rate for the intersection of many prediction sets, which seems to be a promising alternative to Bonferroni. Performance on real datasets is shown.

**Questions:**

1) What are the typical values of $\beta^\ast$ in your numerical experiments? I guess it might be very close to $\alpha$ in many cases as the random forest is one of the predictors and its performance is almost as good as that of the proposed method (Appendices H and I). What about $\hat t$?

2) What are the typical correlations between scores from different predictors in your experiments? In general, the random forest and the XGBoost could provide very similar scores.

3) Overall, LASSO or OLS is not as good as more advanced algorithms, so if we drop them from the ensemble, does it really matter? In other words, under what situations, including such ``simple'' predictors will help? (In terms of coverage and/or size of the resulting prediction set.)

4) Is it possible to establish some proper concentration inequality that quantify the fluctuation of a finite number of random directions $M$?  Or, could you provide some empirical evaluation on the influence of $M$ on coverage and size of the prediction set? (Similar to Table 4.)

**Ethical Concerns:**

["NO or VERY MINOR ethics concerns only"]

**Final Justification:**

The authors propose a theoretically justified method for aggregating multiple valid prediction regions, addressing a timely topic that has received considerable attention recently. The approach is novel and interesting. I believe it substantially advances the research in this direction.

**Limitations:**

Yes.

**Paper Formatting Concerns:**

None.

**Quality:**

3

**Strengths And Weaknesses:**

Strength: It is impressive to see the idea of splitting the calibration set into two (disjoint) parts, with the first part used to calculate a $\beta^\ast$ for coverage correction and the second part for providing a further correction $\hat t$ for constructing the final prediction set. In my opinion, this is the most important contribution of this work to the study of conformal prediction.

Weaknesses:
1) I do not think Theorem 1 precisely covers the setting of Algorithm 1. A complete proof of the correctness of Algorithm 1 must be provided. Theoretical guarantees are perhaps the most valuable feature of conformal prediction. Also, in the statement of the theorem, as well as in its proof (Appendix B) and elsewhere in the paper, the authors should carefully and clearly specify with respect to which random variables the probability is taken. I will raise my rating once these are done satisfactorily.

2) The readability of some parts of the paper needs to be improved, especially on page 5. This is important, as this page contains the most essential idea of the paper. For example, $\mathcal{S}^{K-1}$ is not defined; just above Theorem 1,  what does it mean by ``with this choice of $\{ (u_m, \widetilde{q}_m) \}$''? Does $\widetilde{q}_m$ already contain the information of $\beta^\ast$?

---

> ### Author Rebuttal · Authors · 2025-07-29
>
> We thank the reviewer for their time and questions. We address each of the points successively below.
>
> **A complete proof of the correctness of Algorithm 1 must be provided. Theoretical guarantees…**
>
> We consider the modified form of the theorem statement and proof below, where we make explicit note of the data splitting procedure and the random variables over which the probabilities are defined. In this presentation, we first provide the general theorem and proof for multivariate scores. We then state a corollary for the choice of $T$ (defined in the theorem below) defined with quantile envelopes used in the paper.
>
> **Theorem**
>
> Suppose $(X_1, Y_1), \dots, (X_{N_C}, Y_{N_C}), (X', Y')$ are exchangeable, where $D_C := \\{(X_i,Y_i)\\}_ {i=1}^{N_{C}}$. Assume further that $K$ non-negative maps $s_{k} : \mathcal{X}\times\mathcal{Y}\rightarrow\mathbb{R}_ +$ have been defined and a composite $s(X,Y) := (s_1(X,Y),...,s_{K}(X,Y))$ is defined.
>
> Let $\sigma = (\sigma_1, \dots, \sigma_{N_C})$ be a random permutation of the indices $\\{1, \dots, N_C\\}$, drawn uniformly and independently of $D_C$ and $(X',Y')$. Let the calibration set $D_C$ be partitioned into $D_C^{(1)} := \\{(X_{\sigma_j}, Y_{\sigma_j})\\}_ {j=1}^{N_{C_1}}$ and $D_C^{(2)} := \\{(X_{\sigma_j}, Y_{\sigma_j})\\}_ {j=N_{C_1}+1}^{N_{C_1} + N_{C_2}}$, where $N_{C} := N_{C_1} + N_{C_2}$. Let the corresponding score sets be $\mathcal{S}_ C^{(1)}$ and $\mathcal{S}_ C^{(2)}$. Let $T(\cdot; \mathcal{S}_ C^{(1)}): \mathbb{R}_ +^K \to \mathbb{R}$ be a deterministic function for any given realization of $\mathcal{S}_ C^{(1)}$.
>
> For some $\alpha\in(0,1)$, let $\hat{t}$ be the $\lceil(N_{C_{2}}+1)(1-\alpha)\rceil$-th smallest value of the set of transformed scores $\\{T(s_i; \mathcal{S}_ C^{(1)}) \mid s_i \in \mathcal{S}_ C^{(2)}\\}$. Assume that ties among the transformed scores occur with probability zero.  Then, denoting by $\mathcal{C}(X') = \\{ y \in \mathcal{Y} \mid T(s(X', y); \mathcal{S}_ C^{(1)}) \le \hat{t} \\}$,
>
> $$ \mathcal{P}(Y' \in \mathcal{C}(X')) \ge 1-\alpha, $$
>
> where the probability is defined over the joint draw of the data $D_C$, $(X',Y')$, and the random permutation $\sigma$.
>
> **Proof**
>
> The overall probability is taken over the joint distribution of the exchangeable data, $D_C$ and $(X',Y')$, and the independent random permutation, $\sigma$. We use the law of total probability by first conditioning on a specific realization of the permutation, $\sigma = \pi$, and the data in the first split, $D_C^{(1)} = d^{(1)}$. Given $\sigma = \pi$ and $D_C^{(1)} = d^{(1)}$, the score set $\mathcal{S}_ C^{(1)}$ is fixed. As a result, the function $T(\cdot; \mathcal{S}_ C^{(1)})$ becomes a fixed, deterministic transformation.
>
> By the initial exchangeability of all data points, after conditioning on the values of the first split $D_C^{(1)}$, the remaining $N_{C_2}$ calibration points in $D_C^{(2)}$ and the test point $(X', Y')$ are still an exchangeable sequence. Applying the fixed transformation $T$ to their scores yields an exchangeable sequence of $N_{C_2} + 1$ scalar values:
>
> $$ \\{T(s_i; \mathcal{S}_ C^{(1)}) \mid (X_i, Y_i) \in D_C^{(2)}\\} \cup \\{T(s(X', Y'); \mathcal{S}_ C^{(1)})\\} $$
>
> Under the no-ties assumption, the rank of the test value $T(s(X', Y'); \mathcal{S}_ C^{(1)})$ within this sequence is uniformly distributed on $\\{1, \dots, N_{C_2}+1\\}$. The test point $Y'$ is covered if its transformed score is less than or equal to the threshold $\hat{t}$. This occurs if and only if the rank of the test score is at most $m = \lceil(N_{C_2}+1)(1-\alpha)\rceil$. The probability of this event, conditional on $\sigma=\pi$ and $D_C^{(1)} = d^{(1)}$, is:
>
> $$
> \mathcal{P}(Y' \in \mathcal{C}(X') \mid \sigma = \pi, D_C^{(1)} = d^{(1)}) = \frac{\lceil(N_{C_2}+1)(1-\alpha)\rceil}{N_{C_2}+1} \ge 1-\alpha.
> $$
>
> Since this guarantee holds for any realization $(\pi, d^{(1)})$, the unconditional probability also holds by the law of total probability:
>
> $$
> \mathcal{P}(Y' \in \mathcal{C}(X')) = \mathbb{E}_ {\sigma, D_C^{(1)}}\left[\mathcal{P}(Y' \in \mathcal{C}(X') \mid \sigma, D_C^{(1)})\right] \ge \mathbb{E}_ {\sigma, D_C^{(1)}}[1-\alpha] = 1-\alpha,
> $$
>
> where the expectation is taken over the joint distribution of $\sigma$ and $D_C^{(1)}$. This completes the proof.
>
> **Corollary**
>
> The coverage guarantee of Theorem 1 holds for the $T$ implicitly defined by Algorithm 1. Let $U = \\{u_m\\}_ {m=1}^M$ be the set of projection directions, drawn randomly and independently of the data $(D_C, (X', Y'))$ and the random permutation $\sigma$. Given a realization of the first score split, $\mathcal{S}_ C^{(1)}$, and the directions $U$, the parameters $\\{\tilde{q}_ m\\}_ {m=1}^M$ are determined by the procedure in lines 4-9 of Algorithm 1. The function is then explicitly defined for any score vector $s \in \mathbb{R}_ +^K$ as $ T(s; \mathcal{S}_ C^{(1)}, U) = \max_{m=1,\dots,M} (u_m^\top s / \tilde{q}_ m) $
>
> **Proof**
>
> To prove the corollary, we must show that this specific function $T$ satisfies the conditions of Theorem 1. The overall probability is taken over the joint draw of the data ($D_C, (X',Y')$), the random permutation $\sigma$, and the random directions $U$. We use the law of total probability by conditioning on specific realizations of the random elements $\sigma=\pi$, $D_C^{(1)}=d^{(1)}$, and $U=u$.
>
> Given these fixed realizations, the score set $\mathcal{S}_ C^{(1)}$ and the projection directions $\\{u_m\\}$ are fixed. The procedure in Algorithm 1 to find the base quantiles $\\{\tilde{q}_ m\\}$ via binary search is a deterministic operation on this fixed data. Therefore, the function $T(s; \mathcal{S}_ C^{(1)}, U)$ becomes a fixed, deterministic function of $s$. The conditions of Theorem 1 are met (again assuming no ties in $T$), and its proof implies that the conditional probability of coverage is at least $1-\alpha$:
>
> $$
> \mathcal{P}(Y' \in \mathcal{C}(X') \mid \sigma = \pi, D_C^{(1)} = d^{(1)}, U = u) \ge 1-\alpha.
> $$
>
> Since this guarantee holds for any realization $(\pi, d^{(1)}, u)$, the unconditional guarantee follows from the law of total probability:
>
> $$
> \mathcal{P}(Y' \in \mathcal{C}(X')) = \mathbb{E}_ {\sigma, D_C^{(1)}, U}\left[\mathcal{P}(Y' \in \mathcal{C}(X') \mid \sigma, D_C^{(1)}, U)\right] \ge 1-\alpha.
> $$
>
> Thus, the guarantee holds for the specific procedure in Algorithm 1.
>
> **The readability of some parts of the paper needs to be improved…**
>
> We apologize for this lack of clarity; we were hoping the Appendix A accompaniment would clarify the exposition. For your specific notational comments,  $\mathcal{S}^{K-1}$ denotes the unit sphere in $\mathbb{R}^K$, which we will make explicit where it is introduced in Section 2.2. For the clarification on $(u_m, \tilde{q}_ m)$, **yes**, these parameters are defined once $\beta^\*$ has been determined.
>
> **What are the typical values of $\beta$ and $\widehat{t}$ in your numerical experiments?**
>
> In general, $\beta^\*$ should be fairly close to $\alpha$, and, if the $1-\alpha$ quantile envelopes for $\mathcal{S}_ C^{(1)}$ and $\mathcal{S}_ C^{(2)}$ are similar, $\widehat{t}=1$; $\widehat{t}>1$ if the $\mathcal{S}_ C^{(1)}$ envelope is overly tight and $\widehat{t}<1$ if overly conservative. We show these below for the experiments that were run in the paper. We also highlight that, while Appendix I suggests that the random forest achieved comparable performance to aggregation, this was only true in the simple, UCI baseline experiments. In the more complex settings considered in the main text (Appendix G), our aggregation strategy outperformed the random forest predictor.
>
> $\alpha = 0.05$ -- $\beta^\* = 0.04125$ across all tasks here
>
> | Task ID | $\widehat{t}$ |
> | :--- | :--- |
> | 361234 | 1.093155 |
> | 361235 | 1.818106 |
> | 361236 | 1.272447 |
> | 361237 | 1.575767 |
> | 361238 | 1.172660 |
>
> $\alpha = 0.025$ -- $\beta^\* = 0.020625$ across all tasks here
>
> | Task ID | $\widehat{t}$ |
> | :--- | :--- |
> | 361234 | 1.029694 |
> | 361235 | 1.779467 |
> | 361236 | 1.297137 |
> | 361237 | 1.433094 |
> | 361238 | 1.148173 |
>
> **What are the typical correlations between scores from different predictors in your experiments?**
>
> Here are some examples of correlations of scores from the experiments. As expected, the linear and LASSO models can be highly correlated; CSA is most useful when predictor scores are uncorrelated.
>
> **Task 361234**
>
> $$
> \begin{bmatrix}
> 1& 0.998 & 0.664 & 0.844 \\\\
> 0.998 & 1& 0.672 & 0.856 \\\\
> 0.664 & 0.672 & 1& 0.808 \\\\
> 0.844 & 0.856 & 0.808 & 1
> \end{bmatrix}
> $$
>
> **Task 361235**
>
> $$
> \begin{bmatrix}
> 1& 0.567 & 0.297 & 0.444 \\\\
> 0.567 & 1& 0.25 & 0.301 \\\\
> 0.297 & 0.25 & 1& 0.593 \\\\
> 0.444 & 0.301 & 0.593 & 1
> \end{bmatrix}
> $$
>
> **Under what situations does including such “simple” predictors help?**
>
> You are correct that including such simple predictors should not help. However, we wanted to highlight that the method would not be “thrown off” by the presence of these extra predictors, which we confirm in the ablations for LASSO below. This is because, in a real use case, it may not be a priori clear which predictors are “overly simple” or not.
>
> **Task 361234**
>
> | $\alpha$ | With | Without |
> | :--- | :--- | :--- |
> | 0.05 | 8.0534 | 8.1016 |
> | 0.025 | 9.0303 | 9.0375 |
> | 0.01 | 10.2253 | 10.1544 |
>
> **Could you provide some empirical evaluation on the influence of $M$ on coverage and size of the prediction set?**
>
> Intuitively, as $M\to\infty$, we would expect to recover the $1-\alpha$ tightest cover and, thus, that the prediction region size should be roughly decreasing in $M$, with some plateau. This is borne out below. Across all the choices of $M$, the coverages were identical, namely 0.981 for the $\alpha=0.025$ case and 0.962 for $\alpha=0.05$ (for task 361237).
>
> | $M$ | Length ($\alpha=0.025$) | Length ($\alpha=0.05$) |
> | :--- | :--- | :--- |
> | 50 | 28.037 | 23.017 |
> | 100 | 27.111 | 22.071 |
> | 500 | 25.880 | 22.621 |
> | 1000 | 26.110 | 22.172 |
> | 5000 | 24.943 | 22.037 |

---

> > ### Comment · Reviewer_xX66 · 2025-08-04
> >
> > I am satisfied with the authors' response. The new proofs are clear. I will raise my score later.
> >
> > There is only one point: since only the lower bound of coverage is concerned, there is no need to assume there are no ties. 1) It is possible that there are ties even if the response is continuous (e.g., for tree-based predictors). 2) Usually, the no-ties assumption is introduced to derive an upper bound. It is optional, but the authors are encouraged to establish one if they stick to this assumption.

---

> > > ### Author Response · Authors · 2025-08-05
> > >
> > > We greatly thank the reviewer for raising these additional points. Indeed, as suggested, we can remove the assumption on ties for establishing the lower bound and will do so in the camera-ready version of the paper. We will also include the upper bound proof under the additional ties assumption as suggested by the reviewer.

---

> > > > ### Comment · Reviewer_xX66 · 2025-08-05
> > > >
> > > > Thanks for your efforts. I don’t have any further questions.

---

### Official Review · Reviewer_MGCG · 2025-07-02

**Clarity:** 3
**Significance:** 3
**Originality:** 3
**Rating:** 5
**Confidence:** 4

**Summary:**

Summary:

The paper introduces a novel method called Conformal Score Aggregation (CSA) to enhance uncertainty estimation in ensemble models using conformal prediction. While existing approaches aggregate prediction regions after conformalization, potentially causing overly conservative regions, CSA instead aggregates predictons in the score space itself. The main contribution is using a multivariate score function and defining prediction regions via quantile envelopes. This enables tighter, more data-driven prediction regions. The authors demonstrate that their method achieves distribution-free coverage guarantees. It is less conservative, and improves efficiency in both classification and predict-then-optimize regression tasks through empirical evaluations.

**Questions:**

Questions for Authors:

How does computational complexity scale with increasing ensemble size (K) or calibration dataset size (NC)? Could the method become infeasible at larger scales?

How sensitive is CSA to the choice of projection directions {u_m}? Would alternative direction-sampling methods improve performance further?

How should practitioners optimally select the calibration split ratio (DC^(1):DC^(2)) in practice, and how sensitive are results to variations in this choice?

**Ethical Concerns:**

["NO or VERY MINOR ethics concerns only"]

**Final Justification:**

Initially I mistakenly uploaded review of another paper. And added the review of the submitted paper in the official comment because I was unable to edit the main review at the time. I am able to edit the main review now. Have not received any rebuttal by authors yet on the new review. Will edit this soon after  I receive their rebuttal.

**Limitations:**

yes

**Quality:**

3

**Strengths And Weaknesses:**

Strengths:

The introduction of multivariate score functions and quantile envelopes in conformal prediction is innovative and addresses limitations in existing methods.

The method leverages quantile envelopes and concepts of higher-dimensional quantiles for aggregation, enhancing predictive efficiency.

Authors conduct experimentation, including classification (ImageNet) and regression benchmarks, providing strong evidence of CSA's practical advantages.

CSA can be applied across both classification and regression tasks, unlike some existing methods restricted to specific scenarios.

Weaknesses:

The approach involves multiple stages (splitting calibration sets, directional quantile computation). Authors should comment on the computational overhead, especially with larger ensemble sizes.

The necessity of tuning (e.g., binary search for optimal quantiles) could limit practical scalability.

Performance is likely sensitive to the calibration set split (e.g.,
and
); suboptimal splitting could substantially affect predictive efficiency and coverage.

While the paper extensively compares to previous methods, authors should comment extensively on the computational cost relative to these alternatives.

---

> ### Author Response · Authors · 2025-08-06
> **Updated rebuttal**
>
> We thank the reviewer for their time and questions. We address each of the points successively below.
>
> **How does computational complexity scale with increasing ensemble size (K) or calibration dataset size (NC)? Could the method become infeasible at larger scales?**
>
> The computational complexity is largely the result of the increasing count of directions $M$ one may need in order to capture the complex envelope shapes in higher-dimensional $\mathbb{R}^{K}$ spaces. However, computing the quantile is extremely efficient, since are two primary aspects of the method, (1) computing the envelope threshold $\widehat{\mathcal{Q}}$ and (2) checking whether future points fall into this envelope. Importantly, both of these computations can be vectorized, which makes their computation extremely fast. Please see Appendix D for a full discussion of the vectorization, which highlights how the results can be efficiently computed for even large ensembles.
>
> To demonstrate the computational efficiency of this approach, we reran the experiment on the “Parkinsons” UCI task (from Appendix I) with both varying numbers of predictors ($K$) and projection directions ($M$); for each combination, we measured the total time taken to compute the quantile (i.e. to run Algorithm 1) and to perform the projection to assess coverage for the test points. The additional predictors were taken to be random forests with different numbers of trees. $K$ is given in the left column and $M$ in each column heading, with the entry for each $(K,M)$ pair being reported in seconds.
>
> |$K$|10|100|1000|10000|
> |-:|-:|-:|-:|-:|
> |**6**|0.111668|0.373029|2.32803|37.2561|
> |**8**|0.0961056|0.327051|2.16211|36.7216|
> |**10**|0.117146|0.373464|2.73875|37.2603|
> |**12**|0.123772|0.384527|2.35386|37.0735|
>
> This highlights how the computational cost is minimal and is independent of $K$ when isolated from the directions count.
>
> **The necessity of tuning (e.g., binary search for optimal quantiles) could limit practical scalability.**
>
> We find that the search for $\beta^*$ typically converges within 5-10 iterations, with each iteration being extremely fast to compute, as it is simply assessing for coverage in the fashion described in the previous question. The overall process, therefore, converges within the order of ~10 seconds, and will scale with $M$ according to the same empirically observed rates as those shown in the previous question.
>
> **How sensitive is CSA to the choice of projection directions {u_m}? Would alternative direction-sampling methods improve performance further?**
>
> Empirically, we observed that if the number of directions is large enough, the performance is stable. Intuitively, as $M\to\infty$, we would expect to recover the $1-\alpha$ tightest cover and, thus, that the prediction region size should be roughly decreasing in $M$, with some plateau. This is borne out below. Across all the choices of $M$, the coverages were identical, namely 0.981 for the $\alpha=0.025$ case and 0.962 for $\alpha=0.05$ (for task 361237).
>
> | $M$ | Length ($\alpha=0.025$) | Length ($\alpha=0.05$) |
> | :--- | :--- | :--- |
> | 50 | 28.037 | 23.017 |
> | 100 | 27.111 | 22.071 |
> | 500 | 25.880 | 22.621 |
> | 1000 | 26.110 | 22.172 |
> | 5000 | 24.943 | 22.037 |
>
> Doing alternative direction-sampling may further improve avoid the sampling of redundant directions; however, as discussed in the main paper, the uniform sampling from a hypersphere is a known difficult problem from classical mathematics, which is why this stochastic sampling approach was taken. Nonetheless, an interesting improvement could be a sampling method with faster convergence to Unif($\mathcal{S}$), though the empirical results suggest that this will likely have minimal impact on the end results.
>
> **How should practitioners optimally select the calibration split ratio (DC^(1):DC^(2)) in practice, and how sensitive are results to variations in this choice?**
>
> Empirically, we observed that a split of 20/80 works well empirically, as seen across some of our experimental setups. In general, it is more important to have a large $D_{C^{(2)}}$ to retain the classical upper bound guarantees of conformal prediction that restrict conservativeness. The size of $D_{C^{(1)}}$ largely dictates how accurately the quantile envelope contour can be reconstructed, meaning this becomes more relevant with increasing $K$. In the same way that conformal prediction typically prescribes an 80/20 training/calibration split with practitioners varying around this guideline based on how much training accuracy they are willing to sacrifice or how sample-efficient they believe their model is, we prescribe a 20/80 general split with practitioners having heterogeneous ensembles suggested to have more allocated to $D_{C^{(1)}}$, i.e. a 30/70 split.

---

> > ### Comment · Reviewer_MGCG · 2025-08-06
> >
> > Thank you for the rebuttal. I maintain my positive score.

---

### Official Review · Reviewer_Sni5 · 2025-07-03

**Clarity:** 2
**Significance:** 2
**Originality:** 3
**Rating:** 4
**Confidence:** 2

**Summary:**

This paper aims to improve conformal prediction for ensemble models. While existing work has already proposed aggregating ensemble prediction regions to retain coverage guarantees, this paper introduces a new framework called Conformal Score Aggregation (CSA). CSA leverages quantile envelopes to enable data-driven uncertainty estimation. The method can be directly incorporated into Predict-Then-Optimize tasks. It is evaluated on both regression and classification tasks and demonstrates strong performance.

**Questions:**

Please see above for the weaknesses. All the questions are there.

**Ethical Concerns:**

["NO or VERY MINOR ethics concerns only"]

**Final Justification:**

Most of my concerns are addressed. I will raise my score.

**Limitations:**

yes.

**Paper Formatting Concerns:**

no.

**Quality:**

3

**Strengths And Weaknesses:**

Strengths:

The paper is well-written and well-organized.

The method provides theoretical guarantees for convergence.

It is applicable to both regression and classification tasks.

Weaknesses:

I am not very familiar with this line of work, and I found the paper quite difficult to understand. Section 3.1.1 is especially complex, and I am unsure what the main novelty is and why it is important.

I suggest adding illustrative figures or diagrams to help visualize the process of conformal prediction in ensemble models. This would also help clarify the specific contributions and steps introduced in the paper.

The phrase “improving efficiency” in the title is unclear to me. Does it refer to optimization speed or something else? Clarification would be helpful.

I also found the evaluation results hard to interpret. For instance, what evaluation metrics are used for the regression and classification tasks?

While the abstract mentions uncertainty quantification, the paper does not clearly demonstrate how uncertainty is estimated, nor does it provide an evaluation of the uncertainty quantification quality.

I may improve my score based on further discussion.

---

> ### Author Rebuttal · Authors · 2025-07-29
>
> We thank the reviewer for their time and questions. We address each of the points successively below. Two of the questions are related, so we address these concerns first and then address the remaining ones.
>
> - **The paper does not clearly demonstrate how uncertainty is estimated, nor does it provide an evaluation of the uncertainty quantification quality.**
>
> - **The phrase “improving efficiency” in the title is unclear to me.**
>
> In the conformal prediction community, the goal is to construct prediction regions $\mathcal{C}(x)$ that achieve some pre-determined level of marginal coverage $1-\alpha$, i.e. $\mathcal{P}_ {X,Y}(Y\in\mathcal{C}(X))\ge 1-\alpha$. This is the sense in which the conformal prediction community does “uncertainty estimation”: the replacement of point predictions with $\mathcal{C}(x)$ gives a sense of how that prediction could be wrong (i.e., by any deviation that remains in the prediction region). To make such uncertainty estimates maximally informative, we wish to achieve this coverage guarantee while minimizing the *average size* of such regions, which is referred to as the “predictive efficiency” of the procedure. This is precisely the “efficiency” being referenced in the title. Evaluating different methods of uncertainty quantification, therefore, occurs in two stages: demonstrating the coverage guarantee holds and, if so, then comparing the average prediction region sizes.
>
> **I am unsure what the main novelty is and why it is important.**
>
> As elaborated on in our Related Works discussion, there is now a significant amount of interest in producing these prediction regions for model ensembles. The naive strategy is just to conformalize the ensembled prediction; however, several previous works have demonstrated improvements (in the sense of creating smaller prediction regions) possible by instead conformalizing the models first and then aggregating the individual models’ prediction regions.
>
> These aggregation strategies, however, perform poorly (producing overly large prediction regions) when the uncertainty across models varies significantly. Our method takes a completely different route for aggregation to tackle this issue. Specifically, our core contribution was to define a **multivariate-score** $(s_1(x,y), …, s_K(x,y)$ for an ensemble with $K$ models and to *both* demonstrate that the probabilistic coverage guarantees of conformal prediction could be extended by introducing a multivariate **quantile envelope** in place of the classical **scalar** “threshold” $\widehat{q}$ (from which a partial ordering over $\mathbb{R}^{K}_ +$ could be defined and used to generalize the definition of $\mathcal{C}(x)$) **and** that the resulting prediction regions defined by this procedure were more “efficient” than alternate aggregation strategies, in both classification and regression settings. Extending the coverage guarantees was highly non-trivial and is what is presented in Section 3.1.1, which involved defining a partial ordering using ideas from quantile envelopes and a non-obvious data-splitting step to ensure this partial ordering produces prediction regions that are as tight as possible (which was confirmed to be necessary for coverage in the ablation shown in Section 4.2).
>
> Additionally, using high-dimensional prediction regions in regression settings is often not possible to do directly. One common use case is performing robust decision making downstream of having a conformalized predictor. Section 3.2, therefore, was another significant contribution to highlight that this approach we proposed can be used in decision-making frameworks. This very critically relies on some of the modeling choices of how we constructed the quantile envelopes above and would not trivially follow if these details changed, in particular, in ensuring the constraints can be expressed within the framework of convex optimization, as $u_{m}^{\top} s(\widehat{c}_ {\vec{j}}, c) \le \widehat{q}_ {m} $ $\forall m\in\\{1,...,M\\}$. Notably, *none* of the other aggregation strategies lend themselves to use in this fashion, as they do not take on a similarly explicitly decomposable form, rendering it not possible to efficiently solve the resulting problems using convex solvers.
>
> **For instance, what evaluation metrics are used for the regression and classification tasks?**
>
> As discussed above, the metric of interest in the conformal prediction literature is, assuming a method achieves the desired coverage, how small the average prediction region size is, although how this is measured in regression settings is sometimes non-obvious. In the classification case, one can simply count, for a given $x$, how many of the classes of $\mathcal{Y}$ comprise $\mathcal{C}(x)$. This is what is shown in Table 1: we demonstrate that, for three distinct choices of $\alpha$ (0.1, 0.05, and 0.01), our method attains the theoretically demonstrated coverage guarantee and produces significantly smaller prediction regions than the competitor aggregation strategies and the strategy of conformalizing the ensembled predictor.
>
> In **scalar** regression settings (i.e. $\mathcal{Y} = \mathbb{R}$), prediction region sizes can similarly be assessed by discretizing the output space and checking what subset of such discretized points falls into $\mathcal{C}(x)$. The coverages and region sizes as measured by summing the covered discretized space, therefore, are again what are shown in Table 2, across a wide collection of benchmark tasks, again for two different values of $\alpha$ (0.05 and 0.025).
>
> As mentioned earlier, however, many regression problems are over multi-dimensional output spaces. In these cases, two issues arise. One is that the ability to even estimate the volume of the prediction region (other than some simple analytically computable cases) becomes infeasible due to the exponentially large grids necessary. More importantly, however, it is often not of practical interest to create a discrete collection of points that lie in the prediction region in these high-dimensional cases. These two points, therefore, point to *indirect* measures of the prediction region size as being of greater interest in these high-dimensional settings.
>
> As discussed in the first answer, these high-dimensional regression regions are often useful when employed in robust decision-making pipelines. The standard conformal decision-making framework seeks to solve $w^{\*}_ {\mathrm{rob}} := \mathrm{arg}\min_w \min_{\widehat{c}\in\mathcal{C}(x)} f(w, \widehat{c})$ as a robust proxy to solving the problem under a true, but *unknown*, problem parameter $c$, i.e. $w^\* := \mathrm{arg}\min_w f(w, c)$. In robust decision making, a **larger prediction region** will lead to a **more conservative decision**, which results in a higher suboptimality gap $\Delta(x, c) := \min_{w} \max_{\widehat{c}\in\mathcal{C}(x)} f(w, \widehat{c}) - \min_{w} f(w, c)$. A normalized version of this suboptimality gap is *precisely* the indirect measure that is of interest to compare methods for high-dimensional settings and is what we use to assess and demonstrate improvements in the extremely high-dimensional ($\mathbb{R}^{9867}$) problem of Section 4.3.
>
> **I suggest adding illustrative figures or diagrams to help visualize the process of conformal prediction in ensemble models.**
>
> We were hoping the visual accompaniment provided in Appendix A would clarify the textual description on Section 3.1.1 but would gladly want to extend this to further clarify parts of the method that remain opaque.

---

> > ### Comment · Reviewer_Sni5 · 2025-08-06
> >
> > I think most of my concerns are addressed. I will raise my score.

---

### Decision · Program_Chairs · 2025-09-17

**Decision:**

Accept (poster)

**Comment:**

This paper presents a method for aggregating prediction regions based on  scores produced by individual models in their ensemble. Earlier work merges prediction sets, each of which is generated by each individual model. On the other hand, the current paper considers a vector of scores and leverages quantile envelope to construct a final prediction set. All of reviewers feel that the paper is well written and the method referred to as CSA enhances predictive efficiency. The authors did a good job in responding to reviewers’ comments, leading that two of reviewers raised their overall scores. I believe that this work is one promising research toward conformal prediction for ensembles.